# Critical assessment of metrics and methods used to quantify temporal loading of rainfall events

Molly Asher<sup>1,2</sup>, Mark A. Trigg<sup>1</sup>, Cathryn E. Birch<sup>2</sup>, Rasmus Henriksen<sup>3</sup>, Steven J. Böing<sup>3,4</sup>, and Jonas Wied Pedersen<sup>3,4</sup>

Correspondence: Molly Asher (gy17m2a@leeds.ac.uk)

Abstract. The distribution of rainfall over a storm's duration, known as the event temporal loading, can significantly influence hydrological and geomorphological responses, including run-off generation, urban flood risk, and soil erosion. A wide range of approaches have been developed to analyse rainfall event temporal loading, but these differ in how they characterise rainfall behaviour and in the aspects of storm structure they emphasise. Early research further suggests that climate change may alter rainfall temporal loading in complex and regionally dependent ways, underlining the importance of clear and consistent approaches to its quantification. In this study, we identify 52 metrics that have been applied to describe event temporal loading, and categorise them as classification metrics, summary statistics, or intermittency metrics. We calculate these metrics for 233,128 rainfall events recorded at Danish rain gauges, and demonstrate that, while some metrics are robust to changes in rainfall event temporal resolution and pre-processing, others are highly sensitive. Data-driven cluster analysis further reveals how various metrics relate to one another, highlighting groups of metrics that may be used interchangeably, and others that describe fundamentally different properties. Based on this, we conceptualise five aspects of temporal loading (mass timing, peak timing, magnitude concentration, temporal concentration, and intermittency) and recommend metrics to quantify each. Overall, the study provides a foundation for more deliberate and informed metric selection, helping to align research questions with appropriate representations of rainfall temporal loading, and offering a clearer basis for cross-study comparison.

#### 15 1 Introduction

Rainfall events vary not only in their total volume and duration but also in how rainfall intensities fluctuate as the storm develops. Storms typically evolve through distinct phases of initiation, intensification, and dissipation, reflecting the dynamic interplay of meteorological and cloud processes. In this study, we use the term 'event temporal loading' to describe this internal variability in intensity over the course of a storm. Similar concepts have been described in the literature under various terms, including the intensity profile (Dunkerley, 2021a), storm profile (Kottegoda and Kassim, 1991), rainfall temporal pattern (Wang, 2020), and intra-event rainfall variability (Todisco, 2014).

<sup>&</sup>lt;sup>1</sup>School of Civil Engineering, University of Leeds, Leeds, UK

<sup>&</sup>lt;sup>2</sup>Institute for Climate and Atmospheric Science, University of Leeds, Leeds, UK

<sup>&</sup>lt;sup>3</sup>Weather research, Danish Meteorological Institute. Sankt Kjelds Plads 11, 2100 Copenhagen O, Denmark.

<sup>&</sup>lt;sup>4</sup>DTU Sustain, Technical University of Denmark. Bygningstorvet Building 115, 2800 Kgs. Lyngby, Denmark

30

50

Rainfall event temporal loading has been shown to influence hydrological and geomorphological responses across a wide range of environmental processes, however the direction and magnitude of its effects often vary between studies. For instance, flood modelling experiments have demonstrated that temporal loading alone can produce flood depth differences of up to 35% under identical rainfall volumes (Hettiarachchi et al., 2018). In landslide studies, early-peaking storms have been associated with greater infiltration and more severe slope instability (Fan et al., 2020). Soil erosion studies exemplify the diverging outcomes in temporal loading studies. Wang et al. (2016) reported greater soil loss from later-peaking storms, while Aquino et al. (2013) found early-peaking events to be more damaging. Such differences underscore that while temporal loading is clearly influential, its specific impacts depend on system characteristics, dominant processes, and methodological approaches. Extreme rainfall events are expected to intensify in a warming climate at around 6–7% per degree Celsius, following the Clausius–Clapeyron (CC) relationship (Trenberth et al., 2003). Furthermore, for short-duration convective extremes, some studies suggest even greater rates of intensification, known as super-CC scaling (Fowler et al., 2021). This behaviour has been likely to develop a superior of the process and accuration can be a superior of the process and a convective call develop.

Clausius–Clapeyron (CC) relationship (Trenberth et al., 2003). Furthermore, for short-duration convective extremes, some studies suggest even greater rates of intensification, known as super-CC scaling (Fowler et al., 2021). This behaviour has been linked to dynamical storm changes under higher temperatures, including variations in storm speed, convective cell development, and feedbacks such as latent heat release (Lenderink and Van Meijgaard, 2009; Fowler et al., 2021). These dynamical storm processes have also been suggested to affect the temporal distribution of rainfall within storms. A growing body of work has begun to examine how event temporal loading may change in a warmer climate (Visser et al., 2023; Ghanghas et al., 2024; Asher et al., in press; Wasko and Sharma, 2015), but results to date show variable magnitudes and directions of change, pointing to possible regional dependencies in behaviour. Research in this area remains at an early stage and is developing actively.

Despite its demonstrated importance, rainfall event temporal loading is often oversimplified or overlooked in impact modelling applications. While high-resolution data (e.g., 1–5 minutes) are ideal for capturing the short, intense bursts of rainfall critical for urban hydrology (Schilling, 1991; Einfalt et al., 2004; Berne and Krajewski, 2013), in reality many observational precipitation datasets are provided at lower resolutions. Rainfall impact modelling studies also frequently rely on derived or synthetic events. Here idealised or averaged storm profiles tend to prioritise capturing total volume and duration, at the expense of sub-event dynamics such as peak intensity timing or concentration. For example, symmetrical, centrally peaked intensity profiles are commonly used in flood modelling, such as the FEH design profiles in the UK (Centre for Ecology & Hydrology, 1999) and the Chicago design storm used in the US and other countries (Keifer and Chu, 1957). As a result, the full range of possible hydrological responses may not be represented, leading to underestimation of peak flows, misrepresentation of drainage performance, and failure to identify critical scenarios, compromising risk assessments, infrastructure design, and climate adaptation planning.

Robust, interpretable metrics are therefore essential to bridge the gap between observed rainfall behaviour and the simplified forms commonly used in models. By quantifying key aspects of temporal loading, such metrics help preserve important features when rainfall events are simplified, improving the fidelity of hydrological representations. Furthermore, metrics can offer insight into why the influence of temporal loading varies across studies and contexts, by revealing which specific aspects of storm evolution are most relevant to particular hydrological responses. A wide variety of metrics have been proposed and used to describe event temporal loading. However, there has been little systematic evaluation of how these metrics relate to one another, how sensitive they are to temporal resolution and data processing choices, or whether they are broadly suitable across

hydrological and climatological contexts. There is also a lack of clear terminology: metrics often target different characteristics, such as asymmetry, peakiness, or intermittency, but these distinctions are not always explicit. The relevance of each aspect varies by discipline, and borrowing metrics across domains can result in misplaced emphasis or misinterpretation.

This study seeks to address gaps in knowledge on how temporal loading metrics perform under different data and processing conditions, and on how metrics relate to one another. The work addresses the following research questions:

- RQ1: What key properties of rainfall event temporal loading are commonly measured, and why?
- RQ2: How sensitive are these metrics to the temporal resolution of the rainfall data?
- RQ3: How does de-dimensionalisation of rainfall events affect metric values?
- RQ4: Which metrics are strongly correlated, suggesting they may be redundant or are suitable for use in cross-comparison of studies.

The first research question is addressed through a structured review that identifies existing metrics used to quantify temporal loading, followed by a synthesis of their underlying assumptions, intended applications, and data requirements (Sect. 2). The remaining questions apply data from Danish rain gauges. The second and third questions are examined via comparative analyses of metrics calculated on rainfall events at varying temporal resolutions (Sect. 4.1) and with normalisation applied (Sect. 4.2). Finally, the last research question is tackled using cluster analysis to explore similarities among metrics calculated from raw rainfall events at a 5-minute resolution (Sect. 4.3).

### 2 Literature review

### 2.1 Search and selection strategy

The literature review follows the PRISMA (Preferred Reporting Items for Systematic Reviews and Meta-Analyses) framework (Moher et al., 2010). The PRISMA framework is a standardised method of performing literature review which aims to minimise potential researcher bias, and to promote transparency and reproducibility in the identification and selection of relevant studies. The framework includes four key stages: (1) identification, boolean queries used to identify potential studies, duplicated studies removed; (2) screening, abstract level review to remove irrelevant studies; (3) eligibility, full-text review against pre-defined inclusion/exclusion criteria; and (4) inclusion, relevant studies included for review, and the total number of studies reported.

Boolean queries are search strings using AND, OR and NOT operators, and are constructed here by combining different terms used in the literature to describe event temporal loading, with different terms used to describe rainfall events. The full queries used to search the Scopus, Web of Science, and Google Scholar databases are outlined in Appendix A. The searches were performed on the 15/02/2025 and returned 903 relevant records overall (including 181 on Scopus, 409 on Web of Science, and 313 on Google Scholar). 351 duplicated records were removed, leaving 552 studies for abstract level screening. After screening of abstracts, a further 390 studies were eliminated, leaving 162 papers for full review.

Studies were included if they met the following criteria: (a) use of search terms in the title, abstract, or keywords; (b) published in English; (c) focus on individual rainfall events with data at sub-daily temporal resolution; (d) application of a method or metric to quantify temporal loading. Exclusion criteria were: (a) not published in English; (b) inaccessible or behind paywalls (to the authors, not all included studies are open-access); (c) lack of use of sub-daily rainfall data; (d) combined spatial and temporal analysis (where the temporal loading could not be separated); (e) focus on rainfall trends over extended periods (e.g., seasonal or monthly scales); (f) no application of a metric for quantifying within-event temporal variability.

After applying these criteria, a total of 59 studies remained. We note, however, that several additional studies met all criteria except the use of a metric to quantify temporal loading. These excluded studies fall into three main groups. The first concerns the disaggregation of daily rainfall data (Pampaloni et al., 2021). The second includes stochastic event generation approaches (Cha et al., 2024; Efstratiadis et al., 2022; Kao and Govindaraju, 2008; Nguyen and Chen, 2022; Oriani et al., 2018; Pampaloni et al., 2021). The third involves applications of synthetic rainfall events with prescribed temporal patterns to investigate impacts on soil erosion (De Lima and Singh, 2002; de Lima et al., 2013; Frauenfeld and Truman, 2004; Gao et al., 2018; Parsons and Stone, 2006; Wang et al., 2023), landslides (Zhao et al., 2025), and herbicide runoff (Zhang et al., 1997). In these cases, event profiles are described using broad qualitative labels (e.g. linearly increasing, stepped increasing–decreasing, uniform) rather than being quantified with explicit metrics.

### 2.2 Domains and purposes of reviewed literature

Research on rainfall temporal loading spans a range of domains concerned with hydrological and geomorphological responses to precipitation. The most prominent domain is flood modelling, where studies range from rainfall-runoff models in large watersheds (Cai et al., 2024), to the hydraulic performance of urban sewer systems (Li et al., 2021; Müller et al., 2017). Other domains include soil erosion and sediment transport (Römkens et al., 2002; Liu et al., 2022; Dunkerley, 2021a; Rivera et al., 2012; Gao et al., 2024; Wang et al., 2016; Alavinia et al., 2019; Gholami et al., 2021; Aquino et al., 2013; Klamkowski et al., 2012), pollution modelling (Fu et al., 2021), rainfall-induced landslides (Fan et al., 2020), and climate change impact assessments on extreme rainfall characteristics (Visser et al., 2023; Hu et al., 2018).

Despite the range of contexts, research is generally motivated by similar questions. One common aim is to assess system sensitivity to rainfall temporal loading. This is typically performed by running models with rainfall events of fixed total volume and duration but varying temporal distributions, before examining the resulting impacts on, for example, peak flow, runoff volume, or pollutant wash-off (e.g. Asher et al. (2025)). While these studies treat the system as the subject of analysis (i.e., asking how sensitive a model or process is to rainfall structure), others assume system sensitivity and focus directly on the characterisation of rainfall events themselves. This includes research developing new or more nuanced metrics for describing internal rainfall variability (Todisco, 2014; Dunkerley, 2022), or research applying existing metrics to group and summarise events into representative hyetographs for specific locations, e.g., for China by Wang (2020), Iran by Amin et al. (2000), Slovenia by Dolšak et al. (2016), Greece by Vantas et al. (2019), amongst numerous others. Such work is often intended to improve development of design storms or evaluate how well standard hyetographs reflect observed temporal patterns. Finally, in stochastic rainfall generation and disaggregation, metrics help ensure that synthetic rainfall sequences reproduce realistic intra-

event patterns, preserving statistical characteristics critical for sub-daily modelling. Across all these purposes, the consistent goal is to represent temporal loading more accurately in a way that enhances understanding, modelling performance, and decision-making.

The reviewed studies rely on two main sources of rainfall data: observational records or synthetically generated events. The majority of studies use events extracted from observed rainfall time series, typically recorded by rain gauges, e.g., Zhang et al. (2023), or radar, e.g., Urgilés et al. (2024). Events are extracted from observational records based on assumptions about what constitutes the start and end of a rainfall event, such as minimum intensity thresholds or inter-event dry periods (Restrepo-Posada and Eagleson, 1982). However, event boundaries and characteristics are inevitably shaped by the temporal resolution of the observational dataset, and so 'observed' rainfall events remain approximations shaped by the resolution of the measurement system, rather than exact representations of storm behaviour.

Other studies on event temporal loading use synthetic rainfall events, generated for controlled experimentation or scenario testing (e.g., Cai et al. (2024), Bezak et al. (2018), Fan et al. (2020)). These synthetic events typically fall into two broad categories: idealised design storms and stochastically generated events. Idealised storms take simple geometric forms, such as uniform rainfall, or triangular profiles peaking early, centrally, or late (e.g., Parsons and Stone, 2006), and are used to systematically explore the influence of temporal loading while abstracting from site-specific variability. These types of events are used in the studies noted in Section 2.1 but not included in our review as they do not quantify event temporal loading with a metric. In these studies, the idealised profiles often form the basis for physical experiments, where pre-defined rainfall patterns are applied to laboratory set-ups simulating hillslopes or infiltration systems (Wang et al., 2023; de Lima et al., 2013; Parsons and Stone, 2006; Zhang et al., 1997). In contrast, stochastically generated events, produced using rainfall generators, aim to replicate the statistical properties of observed rainfall while allowing control over event magnitude and duration. The synthetic events are typically applied within modelling frameworks to explore a broader range of possible rainfall scenarios. While use of synthetic rainfall enables controlled investigation of temporal effects, it necessarily involves assumptions about what constitutes a "representative" event. These assumptions can affect the interpretation of results, particularly when attempting to generalise findings to real-world conditions.

### 145 2.3 Rainfall processing choices

150

Research varies not only in the type of rainfall data used, but also in how that data is processed before applying temporal loading metrics. In some studies, metrics are applied directly to the raw rainfall intensity series. This retains the original units and absolute values of the event, allowing metrics to reflect both timing and intensity. In contrast, other studies use double normalised representations, known as dimensionless mass curves (DMCs) or Huff curves (Huff, 1967), in which both cumulative rainfall and event duration are scaled between 0% and 100%. Interpolation is often applied at regular intervals (e.g., every 1% or 10% of the storm duration) to enable comparison between events.

Normalised representations are widely used in studies seeking to generalise across events or derive representative storm profiles. By removing differences in scale and duration, DMCs allow researchers to isolate temporal loading and compare the shapes of events more directly across events of different magnitude and durations. In some cases, mean DMCs are computed

160

for specific regions by averaging across many events (see Sect. 2.2). These regional profiles are used to justify the use of particular design storms or to critique existing design standards. Other studies use DMCs as a basis for classifying events, for instance Huff curves are typically labelled according to the quartile of the Huff curve containing the most rainfall.

Despite the widespread use of DMCs, the effects of normalisation on metric outcomes are often not clearly articulated. Some studies report metrics calculated on both raw and normalised data, but many do not specify the format used at all. This lack of transparency complicates comparisons across studies and makes it difficult to assess whether observed patterns reflect intrinsic features of the events or artefacts of data processing.

### 2.4 Typology of metrics

The wide range of metrics identified in the reviewed studies are listed in Tables C1, C2 and C3 in Appendix C. Based on our own conceptual understanding of the metrics, and their description in the literature, we have assigned each of them to one of three metric types: classification metrics, summary statistics, and intermittency metrics. The categories and how they are delineated is described below.

### 2.4.1 Classification metrics

Classification metrics assign discrete labels to rainfall events. A common approach categorises events by the fraction of the storm that contains the most significant portion of rainfall, typically a third (Horner and Jens, 1942; Aquino et al., 2013; Wang et al., 2016; de Assunção Montenegro et al., 2018; de Andrade et al., 2020), quarter (Huff, 1967; Amin et al., 2000; Jun et al., 2021; Harshanth et al., 2023; Azli and Rao, 2010; Amponsah et al., 2019; Wartalska and Kotowski, 2020; Vantas et al., 2019; Loukas and Quick, 1996; Dolšak et al., 2016; Li et al., 2021; Bezak et al., 2018), or fifth (Villalobos Herrera et al., 2023; Ng et al., 2001; Asher et al., 2025). While simple, the outcomes of such classifications depend heavily on implementation choices. For example, Horner and Jens (1942) introduced the original third-based classification scheme, defining events as advanced, intermediate, or delayed based on the timing of the peak. By contrast, Liang et al. (2023) also use thirds but classify events based on the location of the rainfall mass, identifying the third with the highest total rainfall. Further refinements to this method classify events based on the third containing a summary statistic, e.g. the centre of gravity or D<sub>50</sub> (see Sect. 2.4.2). Figure 1 provides an illustrative example of some classification metrics.

More complex classification schemes compare the dimensionless mass curve of an event to that of a uniform storm. For instance, Terranova and Iaquinta (2011) assign a four-digit Binary Shape Code (BSC) based on whether cumulative rainfall in each quartile exceeds that of a uniform distribution. Kottegoda and Kassim (1991) use the crossing properties concept to classify events with a number based on the number of times the event DMC crosses the uniform DMC, and a letter, which describes whether it exhibits a generally increasing or decreasing intensity profile.

Overall, classification metrics simplify complex event structures into interpretable categories, and provide a tractable way to group storms by structural type. This is particularly useful in design and modelling contexts, where efficiency and consistent processes are important. However, their categorical nature may mask finer differences between events and may be sensitive to how the storm duration is divided.

**Figure 1.** Illustration of some classification metrics for an example rainfall event: (a) the raw rainfall data, and classification of the event based on (b) 3rd with peak (Aquino et al., 2013; Fatone et al., 2021; Pinheiro et al., 2018; Wang et al., 2016), (c) 5th with most rainfall (Villalobos Herrera et al., 2023), (d) 3rd with  $D_{50}$  (Visser et al., 2023).

### 2.4.2 Summary statistics

Summary statistics are continuous metrics that quantify variation in rainfall intensity over time. We broadly group them into three types based on the way they are described in the literature: **peakiness indicators**, **asymmetry indicators**, and **concentration indicators**.

**Peakiness indicators** capture how sharp or pronounced the intensity of peaks are relative to the rest of the storm. Metrics such as *maximum intensity*, *mean intensity*, and *I30* (the maximum rainfall depth accumulated in any 30-minute period) provide baseline measures of peak magnitude and concentration. Other common examples include the *standard deviation*, *coefficient of variation*, *classical skewness*, and *classical kurtosis*, borrowed from general statistics, which reflect variability and the prominence of extremes across the distribution (Brommer et al., 2013; Yang et al., 2015; Kimura et al., 2014; García-Bartual

and Andrés-Doménech, 2017; He et al., 2024). A number of metrics specifically compare the peak with the mean. These include the *peak-to-mean ratio* (also called the rainfall intensity irregularity (Wartalska et al., 2020)), and the *relative amplitude*, defined as the range divided by the mean (Gao et al., 2024; Liu et al., 2022). As the minimum intensity is often 0 mm/hr, and otherwise very close to 0, these two metrics are often equivalent. Similarly, *m2* compares the amount of rainfall in the highest intensity time step to the total event volume (Wartalska et al., 2020), while zone-based metrics, such as the *mean intensity within HIZ* (periods exceeding the event average), quantify how extreme the most active parts of the storm are relative to the overall distribution (Liu et al., 2022).

**Figure 2.** Illustration of several summary statistics for an example rainfall event, including *m1*, *m2*, *m4*, *m5* (Wartalska and Kotowski, 2020), *T25*, *T50*, *T75* (Knighton and Walter, 2016), *centre of mass* and *time to peak* (Jun et al., 2021; Knighton and Walter, 2016)

Lateral asymmetry indicators quantify whether rainfall is concentrated toward the beginning or end of an event. As with classification metrics, some indicators focus on the timing of peak intensity, while others describe the distribution of the rainfall mass across the event duration. Metrics such as *time-to-peak* and the *peak-position-ratio* (also called the coefficient of peak rainfall intensity) describe how early or late the maximum intensity occurs (Fu et al., 2021; Wartalska and Kotowski, 2020; Cai et al., 2024; Wartalska et al., 2020; Knighton and Walter, 2016). *Skew<sub>p</sub>* is also anchored to the peak timing, but measures the relative position of the peak within the rainfall duration (Oh et al., 2024). Other metrics capture asymmetry across the entire rainfall mass. The *m1* metric compares the volume of rainfall before versus after the peak (Wartalska et al., 2020), while the *event loading index* similarly contrasts pre- and post-peak rainfall (Ghanghas et al., 2024). Percentile timing indicators such as the *centre of gravity*, *D*<sub>50</sub> (Visser et al., 2023), and *T25%*, *T50%*, and *T75%* (Knighton and Walter, 2016) describe how

rainfall accumulates over time. Müller et al. (2017) introduce *asymmetry of dependence* to describe the degree to which an event timeseries is reversible. Mass distribution indicators *m3*–*m5* express the proportion of rainfall in the first 33%, 30%, and 50% of the event duration (Wartalska et al., 2020), while Liu et al. (2022) calculate the proportion of rainfall in each quarter of the event (*fraction Q1*, etc). A number of these asymmetry based metrics are visualised in Figure 2.

Concentration indicators describe how rainfall is clustered within an event. A key example is the *Precipitation Concentration Index (PCI)*, originally developed to assess monthly and seasonal rainfall distributions (Oliver, 1980) but also applied to individual events (He et al., 2022). Higher *PCI* values indicate a greater share of rainfall concentrated in fewer time steps. The *Temporal Concentration Index (TCI)* was introduced as a scale-independent alternative that could be used to compare across events of different durations (Long et al., 2021). It quantifies the deviation of an event's temporal loading from that of a uniform rainfall event with the same duration and volume. Another related measure is the normalised root-mean-square error around the peak (*NRMSE<sub>p</sub>*), which assesses how tightly rainfall is clustered near the peak time step (Oh et al., 2024). The *Gini coefficient*, originally developed to measure income inequality, has also been adapted to assess the unevenness of rainfall distribution within an event (Lu et al., 2025). Applied to rainfall, it reflects how much cumulative rainfall deviates from a perfectly even distribution. For highly concentrated events, the accompanying *Lorenz asymmetry coefficient* describes whether this arises from a few extreme high-intensity bursts or from many moderately intense intervals, both of which could lead to high concentration. Zone based indicators, including the % *of rainfall in high-intensity zone (HIZ)*, *time in high-intensity zone (HIZ)/low-intensity zone (LIZ)*, indicate how long the storms stays intense and how much rainfall is concentrated in these intense bursts (Liu et al., 2022).

### 2.4.3 Intermittency metrics

Intermittency metrics are designed to capture the presence and frequency of dry periods within rainfall events. These metrics are particularly relevant in contexts where wet–dry fluctuations affect hydrological processes such as infiltration, soil saturation, and flash flood generation. Although fewer in number than other metric types, intermittency measures add an important dimension to the characterisation of temporal loading by highlighting fluctuations that may not be captured by peak-based or cumulative statistics. The *event-dry ratio* calculates the proportion of time steps within an event that record no rainfall as a proportion of the total number of recorded intervals in the event (Pohle et al., 2018). Relatedly, the *intermittency fraction* measures the number of transitions from wet to dry during the event, giving a more specific idea of whether event rainfall is very disparate (Dunkerley, 2021b).

### 2.5 Metrics summary

Sects. 2.2–2.4 review the selected studies in terms of their objectives, the types of metrics employed, the rainfall data sources used, and the processing steps applied. The distribution of these characteristics across the reviewed studies is summarised in Table 1, while a full overview of all studies is provided in Table B1 in Appendix B.

Figure 3 presents the number of studies applying each metric and highlights the geographical focus of those studies. The figure shows that most metrics have only been applied in one or two papers, with a small number dominating the literature.

**Table 1.** Summary of studies by purpose, metric type, the type of rainfall events used in the study, and the processing stages applied. Note that some studies may be counted under more than one categorisation

| Category                     | Subcategory                               | Detail             | Number of studies |
|------------------------------|-------------------------------------------|--------------------|-------------------|
|                              | Characterise rainfall profiles in an area | 23                 |                   |
|                              |                                           | Flooding/hydrology | 12                |
|                              |                                           | Landslide          | 1                 |
| Study purpose                | Impact of storm temporal profile on:      | Pollution          | 2                 |
|                              |                                           | Soil erosion       | 13                |
|                              | Climatic study                            |                    | 5                 |
|                              | Stochastic event generation               |                    | 3                 |
|                              | Evaluating satellite data                 |                    | 1                 |
|                              | Metric development                        |                    | 4                 |
| N/14** 4                     | Classification                            |                    | 33                |
| Metric type                  | Summary stats                             |                    | 25                |
|                              | Intermittency                             |                    | 2                 |
| D - 1 - 6 - 11 4             | Real                                      |                    | 47                |
| Rainfall type                | Synthetic                                 |                    | 12                |
| ) a i u f a 11 a a a a i a a | Raw rainfall events                       |                    | 24                |
| Rainfall processing          | Dimensionless Mass Curves                 |                    | 28                |
|                              | Unclear                                   |                    | 7                 |

Classification metrics, which have been in use for several decades, are the most prevalent. Among these, two interpretations of Huff quartiles, the *4th with most* and *4th with peak*, first introduced in 1967, have been applied extensively across different regions. The *3rd with peak*, derived from the earlier work of Horner and Jens (1947), is also frequently used, with a particular concentration of studies focusing on soil erosivity in Brazil.

#### 250 3 Methods

### 3.1 Rainfall data

This research uses rainfall events extracted from Danish rain gauge data at 1 minute resolution. Figure 4 shows the national Danish rain gauge network, which is maintained and operated by the Danish Meteorological Institute (DMI). The individual gauges in the network are mainly owned by DMI and the Danish water utilities organised through the 'Water Pollution Committee of The Society of Danish Engineers' (abbreviated 'SVK" in Danish for 'Spildevandskomiteen'). DMI's gauges are

**Figure 3.** Distribution of metrics across the reviewed studies, showing both the number of studies in which each metric was applied and the countries from which these studies were drawn.

weighing rain gauges of the brands Geonor and OTT Pluvio<sup>2</sup>. The measurement resolution of Geonor gauges are 0.1 mm, while the resolution of the Pluvio<sup>2</sup> gauges started at 0.1 and was updated during the time period focused on in this study to 0.01 mm. SVK's gauges are tipping bucket gauges by Rimco with a measurement resolution of 0.2 mm. Both gauge types record data with a temporal resolution of one minute. Figure 4 (left) shows that all DMI's gauges have been operational for less than 15 years, while there is a much larger spread in time series lengths for the SVK gauges with the oldest ones providing continuous time series from 1979 to the present. Figure 4 (right) shows the spatial locations of the gauges with DMI gauges being relatively evenly spread across the country, while SVK gauges cluster around the major cities.

Data quality control for all gauges in the network are performed manually by DMI's climatology department (DMI, 2025). Data points ruled non-trustworthy by manual quality control are excluded from this study.

# 3.2 Rainfall event extraction and pre-processing

For each rain gauge, independent rainfall events are extracted over the full period of available data. The rainfall time series is first aggregated to 5-minute resolution. This resolution is deemed to best preserve temporal detail, while minimising noise due to the measurement resolution of the rain gauge data. This noise arises because both tipping-bucket gauges and weighing gauges record rainfall in discrete increments, when the buckets tip or when the accumulated weight flips a decimal, which leads to many small, artificial peaks in the 1-minute time series. To ensure event independence, we extract events using a minimum interevent time (MIT) threshold (Restrepo-Posada and Eagleson, 1982; Molina-Sanchis et al., 2016). An 'event' thus constitutes

**Figure 4.** A stacked histogram of the length of the time series for rain gauges (left), and a map of Denmark showing the locations of each gauge, including indications of gauge ownership and time series length (right). Base map: © OpenStreetMap contributors 2025. Distributed under the Open Data Commons Open Database License (ODbL) v1.0.

any rainfall separated by at least 11 hours of rain-free conditions, following practice in several Danish hydrological studies (Gregersen et al., 2013; Thomassen et al., 2023). This approach ensures that each event begins and ends with non-zero rainfall. The MIT parameter has been shown to play an important role in determining the number and properties of rainfall events selected (Dunkerley, 2008). For temporal loading, which is interested in what happens around the peak, the way in which the edges of the event are defined is particularly important, but in this research we do not investigate its influence further.

We define and analyse entire rainfall events, rather than only the most intense 'burst' periods, capturing the complete temporal evolution of each storm. Events with less than 4 mm of total rainfall are excluded to remove very light events unlikely to be hydrologically significant. This process produces a dataset of 233,128 rainfall events observed between 1979 and 2025.

Coarser-resolution versions of each event are also derived at 10-, 30-, and 60-minute resolutions. These are generated based on the start and end times of the 5-minute event, so for a given coarser resolution, the first timestamp equal to or before the 5-minute event start is taken as the new start, and the last timestamp equal to or later than the event end is taken as the new end. As a result, the duration of events at coarser resolutions may be equal or longer in the duration than their 5-minute counterparts.

### 3.3 Dimensionless mass curve generation

Dimensionless mass curves (DMCs) are generated from all 5-minute resolution rainfall events. A DMC represents the cumulative distribution of rainfall within an event, scaled such that both time and accumulated rainfall range from 0% to 100%. Each DMC is interpolated to have 10 equally spaced time points. Importantly, while DMCs are typically defined as cumulative profiles, in this analysis we derive a double normalised incremental representation by converting each interpolated DMC back into incremental rainfall. This version reflects rainfall intensity as a proportion of total event mass, distributed over relative time.

All temporal loading metrics in this study are applied to these 10-point, double normalised incremental series rather than to cumulative curves. This preserves consistency with how metrics are typically applied to raw rainfall intensities, while enabling direct comparison across events in a dimensionless domain.

### 3.4 Metric computation and post-processing

Rainfall temporal loading metrics are implemented in Python based on the definitions provided in the original publications. In some cases, the literature lacks sufficient detail on the precise application of metrics (e.g., with respect to temporal resolution, handling of zero values, normalisation procedures). Where ambiguity exists, we make reasonable interpretations of the provided formulations. Two of the identified metrics - the Binary Shape Code and the event's Crossing Properties - are not straightforward numeric indicators and are thus deemed outwith the scope of this study.

In addition to the literature-identified metrics, we compute a set of statistical moments on each event's intensity profile to help interpret our clustering results. These include classical moments, which are calculated on the raw sequence of intensities without respect to their temporal order. These include the *classical mean*, which is the mathematical average of the intensity values; the *classical standard deviation*, which is the root-mean-square deviation of intensities from their mean, i.e. the typical spread or variability of rainfall rates (previously applied to rainfall events by García-Bartual and Andrés-Doménech (2017)); the *classical skewness*, which measures the asymmetry of the intensity distribution around the *classical mean*; and the *classical kurtosis*, which measures tail-heaviness or peakiness of the intensity distribution around that centre.

We also calculate temporal moments which consider the distribution of intensity values in respect to time. These include the *temporal mean* (referred to here as the *centre of gravity*), which weights each time step by its rainfall intensity to calculate the average time at which rainfall occurs during the event, normalised between 0 and 1; the *temporal standard deviation*, which measures the spread of rainfall in time around this temporal mean, indicating how temporally concentrated or dispersed the rainfall is over the event duration; the *temporal skewness*, which measures the asymmetry of rainfall in time around this *temporal mean*; and the *temporal kurtosis*, which quantifies how sharply the rain is clustered around that centre.

The full list of metrics applied, including those from the literature alongside additional statistical and temporal moments, is presented in Table 2. The full codebase for rainfall event extraction and metric calculation is openly available at our GitHub repository (https://github.com/masher92/MetricEvaluation/), providing a consistent and reproducible framework for future applications of these metrics.

# ${\bf 3.5}\quad Testing\ sensitivity\ to\ temporal\ aggregation\ (RQ2)$

To assess the sensitivity of each metric to temporal aggregation of rainfall data, we compare metric values computed on rainfall events derived from data aggregated at different temporal resolutions. We treat metrics computed on events from 5-minute rainfall data as the reference ('truth') and compare them to values obtained from coarser aggregations at 10-, 30-, and 60-minute intervals.

Table 2. List of metrics within each typology applied in this research

| Metric type                  | Metrics                                                                          |  |  |  |
|------------------------------|----------------------------------------------------------------------------------|--|--|--|
| Classification               | 3rd / 4th / 5th with peak;                                                       |  |  |  |
|                              | 3rd/4th/5th with most;                                                           |  |  |  |
|                              | 3rd with $D_{50}$ ; 3rd with $CoG$ .                                             |  |  |  |
| Summary statistics           | Max intensity; 130; Cv; (C) skewness; (C) kurtosis; mean intensity; (C) std; (T  |  |  |  |
|                              | skewness; (T) kurtosis; (T) std; Centre of gravity; Peak-position ratio; time to |  |  |  |
|                              | peak; Peak-mean ratio; Relative amplitude; Skewp; NRMSEp; Gini coefficient;      |  |  |  |
|                              | Lorenz asymmetry coefficient; Event loading index; Fraction of rainfall in Q1 /  |  |  |  |
|                              | Q2 / Q3 / Q4; m1; m2; m3; m4; m5; T25; T50; T75; PCI, TCI; Mean intensity        |  |  |  |
|                              | HIZ; % time in LIZ / HIZ; % Rain in HIZ.                                         |  |  |  |
| <b>Intermittency metrics</b> | Intermittency fraction; Event-dry ratio.                                         |  |  |  |

For each metric and resolution pair, we quantify both numerical and ranking sensitivity. These two complementary measures allow us to distinguish between metrics which exhibit substantial changes in raw values (numerical sensitivity), those which show shifts in the relative ordering of events (ranking sensitivity), and those which display both or neither aspect of sensitivity.

For continuous metrics, the numerical sensitivity is assessed using the symmetrical median absolute percentage error (sMAPE)

(Makridakis and Hibon, 1995). sMAPE measures the absolute percentage error between the metric values at 5 minute and at coarser resolutions, and is calculated as:

$$sMAPE = \frac{100}{n} \sum_{i=1}^{n} \frac{|x_i - y_i|}{(|x_i| + |y_i|)/2}$$
(1)

where  $x_i$  and  $y_i$  are the original and transformed metric values for event i, respectively.

The ranking sensitivity is assessed using Spearman's rank correlation coefficient ( $\rho$ ), which quantifies the degree to which the relative ordering of events changes due to temporal aggregation, and is calculated as:

$$\rho = 1 - \frac{6\sum_{i=1}^{n} d_i^2}{n(n^2 - 1)} \tag{2}$$

where n is the number of paired observations,  $d_i = R(x_i) - R(y_i)$  is the difference between the ranks of the i<sup>th</sup> pair of observations.

For categorical metrics, the % of events in a different category from 5 minute at each coarser resolution is used for numerical sensitivity, and Kendall's  $\tau$  quantifies the ranking sensitivity (Kendall, 1938) by measuring the association between two ranked variables. It is defined as:

345

$$\tau = \frac{P - Q}{\sqrt{(P + Q + T)(P + Q + U)}}\tag{3}$$

where P / Q are the number of concordant / discordant pairs, and T is the number of ties only in x, and U is the number of ties only in y. If a tie occurs in both x and y for the same pair, it is not counted in either T or U.

We also visually explore how each metric responds to aggregation by plotting distributions across resolutions (using histograms for continuous metrics, and bar plots for categorical metrics).

### 3.6 Testing sensitivity to normalisation (RQ3)

To evaluate the robustness of rainfall metrics to dimensionless mass curve (DMC) conversion, we compare values calculated on raw 5-minute rainfall events and DMC-transformed rainfall time series. Note that, I30 cannot be calculated, as DMCs have no concept of 30-minutes. Likewise the Frac. in Q1/2/3/4 can only be calculated on events with n timesteps divisible by 4, and so are not calculated either. Following the methods for temporal aggregation sensitivity testing (Sect. 3.5), we calculate numerical and ranking sensitivity of continuous metrics using sMAPE and Spearman's p, and for categorical metrics using % differences in categorisation and Kendall's  $\tau$ . Metric value distributions are also compared using histograms for continuous metrics, and bar plots for categorical metrics.

#### 350 3.7 Testing metric redundancy and complementarity (RQ4)

To identify metrics which are strongly correlated and which may therefore be considered redundant or interchangeable, we apply agglomerative hierarchical clustering based on pairwise metric similarity across all events (Jain et al., 1999). Similarity between metrics is quantified using the absolute Spearman rank correlation coefficient, and clustering is performed using average linkage. While clustering directly on metric vectors using Euclidean distance was considered, this would have prioritised numerical proximity over functional similarity. This is problematic because some metrics which have very high negative correlation, and likely describe the same feature, are very distant in Euclidean space. Prior to clustering, each metric is assessed for skewness, and those with an absolute skewness 

### 4 Results and discussion: Quantitative metric analysis

#### 4.1 Sensitivity to temporal aggregation (RQ2)

The sensitivity of each temporal loading metric to changes in the temporal resolution of the rainfall data is examined in Figure 5. Metrics located in the bottom-right corner of their plot have high robustness (low numerical and ranking sensitivity), whereas those trending upwards and to the left show increasing sensitivity to aggregation. These results are supported by Figure 6, which shows histograms of metric values across all events, with overlain distributions for each temporal resolution. Robustness varies substantially, pointing to meaningful patterns in behaviour.

Metrics that describe the peak intensity of rainfall events consistently display a low numerical robustness to temporal aggregation in Figure 5. This includes metrics such as the *peak-mean ratio* and *relative amplitude*, which compare the magnitude of the peak to the overall average intensity. As the temporal resolution becomes coarser, these metrics are distorted because, while the event mean remains relatively stable, the peak intensity is often smoothed, causing the relative peak intensity to decrease. This effect is visible in Figure 6, where distributions for the *relative amplitude*, *peak-mean ratio* and *maximum intensity* shift towards lower values at coarser resolutions. Low numerical robustness is also seen in Figure 5 for m2, the % of rainfall in the whole event found in the peak time-step. However, in this case, the distribution in Figure 6 shifts in the opposite direction, towards higher values. This reflects coarser aggregations having larger time-steps, meaning that the highest-intensity interval captures a larger fraction of rainfall. While these peak intensity-based metrics exhibit high numerical sensitivity to temporal aggregation, their ranking sensitivity remains comparatively low, as evidenced by Spearman's  $\rho$  values exceeding 0.8 for all metrics discussed. This indicates that although absolute values change with resolution, the relative ordering of events is largely preserved, suggesting that such metrics may still be useful for comparative analyses, even when temporal resolution varies.

Metrics that describe the timing or position of the peak, such as time to peak and peak-position-ratio, or that are anchored to the timing of the peak, such as m1, event loading index and  $skew_p$ , are also sensitive to resolution. In contrast to the peak intensity metrics, however, these timing-based metrics show both high numerical and ranking sensitivity. This is expected, as temporal aggregation not only smooths peak intensities but can also displace the perceived timing of the peak. In particular, where the aggregation process assigns aggregated values to the end of time intervals, the peak can be effectively shifted later in time at coarser resolutions. Figure 6 shows broader, more variable distributions for these metrics under coarser resolutions, reflecting increased distortion. A similar pattern holds for categorical metrics. Those that classify events based on the fraction containing the peak rainfall (3rd with peak, 5th with peak) exhibit greater sensitivity to resolution changes than those based on the fraction containing the bulk of rainfall (3rd with most, 4th with most, 5th with most). This suggests that metrics tied closely to the peak are generally more resolution-dependent, while metrics reflecting broader event structure tend to be more robust.

The statistical moments *classical skewness* and *classical kurtosis* both show high numerical and ranking sensitivity to temporal aggregation in Figure 5. As seen in Figure 6, their distributions shift leftward at coarser resolutions, suggesting that rainfall events appear more symmetrical and less heavy-tailed. This reflects the mathematical sensitivity of these moments to extreme values and their positions within the distribution, features that are especially affected by aggregation. Smoothing sharp, intense

**Figure 5.** Sensitivity of metrics to temporal aggregation. For each metric (separate subplot), points represent comparisons between the metric calculated at coarser resolutions (10-, 30-, 60-minutes) and the reference 5-minute data. For (a). continuous metrics (summary statistics and intermittency metrics), the x-axis shows Spearman's rank correlation coefficient (p) and y-axis symmetrical mean absolute percentage error (sMAPE) relative to the 5-minute baseline. For (b). categorical metrics, the x-axis shows Kendall's tau, and y-axis shows the percent disagreement with 5-minute baseline.

peaks reduces the extremity of the tails (lowering kurtosis), while averaging over longer intervals can flatten sharp rises or falls in intensity, reducing asymmetry in the distribution of intensity values (lowering skewness). Ranking sensitivity is also high, as aggregation can reorder events by altering the prominence of subtle asymmetries or outliers.

Metrics that split events into high- and low-intensity zones (HIZ / LIZ) show moderate ranking sensitivity, but fairly low numerical sensitivity (Figure 5). The distribution of % time in HIZ shifts right at coarser resolutions, while for % time in LIZ it shifts left. A higher proportion of time above the mean intensity at coarser resolutions suggests that fewer time steps fall below the mean when extremes are smoothed (and the mean is lower). The % rainfall in HIZ shows less change across resolutions. Its distribution is already heavily skewed toward high values, and aggregation slightly lowers the upper tail, suggesting fewer extreme values, but does not substantially alter the metric's central tendency. The mean intensity in HIZ is relatively robust across resolutions.

The highest robustness in Figure 5 is observed for the time-based moments, including *temporal skewness*, *temporal kurtosis*, *temporal std*, and *centre of gravity*, and percentile timing metrics like  $D_{50}$  and T75. These summarise the overall timing and distribution of rainfall across the event, which is less affected by resolution changes than short-lived peaks. Their distributions in Figure 6 remain tightly clustered and largely unchanged across resolutions. Categorical metrics based on these values, such as the 3rd with  $D_{50}$ , or the 3rd with CoG, also show the greatest stability among categorical approaches.

Together, Figures 5 and 6 indicate that metrics capturing the cumulative structure and timing of rainfall events can be used with confidence even when only coarse resolution data are available. By contrast, those reliant on the timing or magnitude of short-lived intensity bursts should be applied with caution in such cases.

### 4.2 Sensitivity to normalisation (RQ3)

Sensitivity to conversion of raw 5-minute rainfall into double-normalised, 10-step representations (DMCs) varies substantially across temporal loading metrics, as shown in Figure 7 and Figure 8. As with temporal aggregation (Section 4.1), metrics that describe the overall distribution of rainfall over time show the least sensitivity to normalisation. In Figure 7a, these robust metrics cluster in the bottom-right and are shown in more detail in the inset. They include mass distribution indicators (m3, m4, m5), percentile timing metrics (T25, D/T50, T75), and moment-based metrics ( $centre\ of\ gravity$ ,  $temporal\ skewness$ , and  $temporal\ kurtosis$ ). The value distributions for these metrics in Figure 8 also remain largely unchanged after DMC conversion. These metrics are inherently robust to both interpolation and loss of dimensionality because they reflect the shape and balance of the event over its duration, rather than relying on specific peaks or absolute magnitudes. Their values are either unitless by construction or meaningful when expressed as proportions of total event time or mass, making them more stable across transformations. Categorical metrics based on these values, such as the  $3rd\ with\ D_{50}$ , or  $3rd\ with\ CoG$ , as well as those recording the fraction with most rainfall ( $3rd\ /\ 4th\ /\ 5th\ with\ most$ ), also show the greatest stability among categorical approaches (Figure 7, Figure 8).

By contrast, several metrics that take continuous values on raw rainfall series become discretised when applied to DMCs interpolated to only 10 time steps. These include % time in LIZ / HIZ, event dry ratio, intermittency, peak-position-ratio, centre of gravity, skew<sub>p</sub>, and classical skewness. These depend on the proportion of event time spent above or below certain

**Figure 6.** Sensitivity of metric distributions to temporal aggregation. Each subplot shows how the distribution of a given metric changes when calculated on rainfall data at different temporal resolutions (5-, 10-, 30-, 60-minutes). For (a). continuous metrics, this is shown using overlaid histograms. For (b). categorical metrics, grouped bar charts display the relative frequency of each category across resolutions.

435

440

**Figure 7.** Sensitivity of metrics to dimensionless mass curve representation. For each metric, points represent comparisons between the metric calculated with 5-minute raw data and with a 10-point, DMCs calculated on 5-minute data. For (a). continuous metrics (including summary statistics and intermittency metrics), the x-axis shows Spearman's rank correlation coefficient (*p*) and y-axis the symmetrical mean absolute percentage error (sMAPE). For (b). categorical metrics, the x-axis shows Kendall's tau and the y-axis the percent disagreement between the pairs of metric values.

thresholds or in specific zones, and so are limited in the DMC to multiples of 10%, constraining meaningful variation in range and granularity across events. This is very apparent in the stepped histograms of these metrics in Figure 8a Similarly, the interpolation used in the DMC conversion introduces a form of resolution coarsening similar to temporal aggregation. As with coarser time steps, this process smooths sharp transitions and reduces variability, disproportionately affecting metrics that are sensitive to peak timing and magnitude, such as *peak-mean ratio*, *relative amplitude*, *peak-position-ratio*, *m1*, *m2*, and the *event loading index*.

However, the effects of DMC transformation extend beyond temporal smoothing. Unlike temporal aggregation, DMC conversion involves full de-dimensionalisation, with both time and intensity rescaled to a [0,1] domain. This has more severe consequences for metrics that rely on real physical quantities. For instance, *mean intensity* becomes meaningless, as the total intensity is fixed to 1, and so the mean intensity is 0.1 in all cases (Figure 8a). Similarly, metrics that rely on absolute intensity contrasts, such as the *standard deviation* or *coefficient of variation*, lose interpretability when all intensities are normalised. In these cases, relative differences are compressed and the metric no longer reflects the original signal.

**Figure 8.** Sensitivity of metric distributions to double normalisation in DMCs. Each subplot shows how the distribution of a given metric changes when calculated on raw rainfall data at 5-minute temporal resolution and on a double-normalised version of the event, interpolated to ten data points. For (a). continuous metrics, distributions are shown using overlaid histograms. Note that gaps in some histograms arise from the discretisation of values when metrics are computed on DMCs—certain metric values are simply not possible, so some histogram bins remain empty. For (b). categorical metrics, grouped bar charts show the relative frequency of each category across resolutions.

This distinction is particularly important when DMCs are used to generate synthetic events or inform design storms. In these cases, dimensionless curves are often re-scaled using external information about storm depth and duration to reconstruct plausible, physically realistic events. Metrics that remain interpretable in the dimensionless domain are therefore better suited to characterising DMC shapes or classifying rainfall types, while those that depend on physical units or fine-grained temporal variation may require adaptation or caution when applied to normalised forms.

Taken together, these findings underscore the importance of considering input representation when comparing rainfall met450 rics across studies, particularly when simplifications like DMCs are involved. Metrics that are robust to DMC transformation
may be more appropriate for comparative analysis or pattern identification, while others may require adjusted interpretation
when removed from their physical context.

### 4.3 Metric redundancy and complementarity (RQ4)

Figure 9a plots a dendrogram which visualises the results of hierarchical cluster analysis based on the similarity between each pair of temporal loading metrics across the rainfall event dataset. The horizontal distance between each metric and where it joins with another branch represents similarity, with items merging at shorter distances being more similar. Twelve clusters are used, which was indicated in silhouette score analysis to be the optimal number to increase in-group cohesion, and reduce between-group separation (see Figure D1, Appendix D). On the dendrogram, the clusters containing more than one metric are coloured and labelled with a cluster number. The resulting clusters reveal groups of metrics that behave similarly across events and may therefore capture overlapping or redundant structural information.

Cluster 1 is the largest cluster, with fifteen metrics, and relates to the timing of mass distribution. This includes classification metrics (3rd with most, 4th with most, 5th with most, 3rd with  $D_{50}$  and 3rd with CoG), as well as percentile-based metrics (T25, T50, T75), mass-distribution indicators (m4, m3, m5), temporal skewness, the centre of gravity, and the fraction of rainfall in Q1. This cluster contains metrics with the strongest (positive or negative) correlations, as can be seen in Figure 9b which plots the Spearman's correlation matrix. Negative correlations arise where metrics respond to the same underlying asymmetry but with reversed directionality. For instance, a front-loaded event will have a lower  $D_{50}$ , but a higher m3 (proportion of rainfall in the first third). The high correlations between metrics in the cluster suggests that these metrics can be used interchangeably to rank the asymmetry in rainfall mass distribution in a set of rainfall events.

Cluster 2 contains a grouping of eight metrics clearly related to the timing of the peak timestep. Metrics in this peak timing cluster include several classification metrics (e.g. 3rd with peak, 4th with peak, 5th with peak, and 3rd (D<sub>50</sub>)), as well as summary statistics that explicitly quantify peak position (e.g., peak-position-ratio, time-to-peak, and skew<sub>p</sub>). Interestingly, the metric m1, representing the ratio of rainfall before versus after the peak, is also placed within this cluster. Although m1 is seemingly focused on the balance of rainfall mass distribution rather than peak timing, its position within this cluster suggests that, in practice, it is strongly influenced by peak placement. Cluster 2 appearing adjacent to, but distinct from, cluster 1 implies that while both peak and mass timing relate to the temporal distribution of rainfall, they are not completely interchangeable concepts. This highlights the importance of establishing the aspect of temporal loading most relevant to a specific application before selecting metrics to utilise.

480

485

**Figure 9.** (a) Dendrogram visualising results of hierarchical cluster analysis of the pairwise similarity of metrics across all events. Similarity is quantified using the absolute value of Spearman's  $\rho$ , and clustering is performed using agglomerative hierarchical clustering with average linkage and 12 clusters. Clusters with more than one member are highlighted in red and marked with a cluster number. (b) The full Spearman correlation matrix upon which clustering is based

Unlike the first two clusters, which both capture aspects of when rainfall occurs within the event, whether in terms of the distribution of mass or the relative timing of the peak, the metrics in the third, fourth and fifth clusters are not concerned with sequencing in time. Instead, they describe the magnitude of the peaks and how rainfall is concentrated among timesteps (magnitude concentration), and for some metrics, how closely such high-intensity timesteps occur (temporal concentration), without distinguishing whether these bursts fall early, mid, or late in the event.

The third and fourth clusters contain metrics relating to the peak magnitude and magnitude concentration. In cluster 3, the first of the three subclusters identified on the dendrogram is formed by *maximum intensity* and *I30* (the maximum intensity in any 30-minute period). These both capture the magnitude of the single wettest short-duration period without reference to the rest of the event. The second subcluster groups the *mean intensity in the HIZ* (timesteps above the event's mean intensity) with *classical standard deviation* (the variation of rainfall intensities around the mean intensity). Both metrics emphasise how rainfall is distributed relative to the event mean, reflecting the degree to which rainfall is concentrated into particularly intense phases as opposed to being spread more evenly across the event. The third subcluster contains *m2* (percentage of total rainfall

500

505

520

in the highest-intensity timestep), *PCI*, and *NRMSE<sub>p</sub>*. These metrics explicitly compare rainfall in some timesteps with rainfall in others, directly quantifying how concentrated rainfall is within an event. They also share a potential sensitivity to event length, tending to return higher values when an intense burst is embedded in a short event than when it appears in a longer event with more low-intensity timesteps. Taken together, the three subclusters demonstrate that cluster 3 relates to metrics of storm magnitude and concentration. The first subcluster isolates the absolute peak intensity, while the second and third reflect different ways of capturing how rainfall is concentrated across timesteps. Events with very high peaks often also display strong contrasts between high- and low-intensity periods, which is consistent with the behaviour of concentration-focused metrics such as *mean intensity in the HIZ*, (*C*) *std*, *m2*, *PCI*, and *NRMSE<sub>p</sub>*.

Cluster 4 relates to magnitude concentration, but focuses more on metrics that reflect distributional imbalance, showing how rainfall is unevenly shared across timesteps rather than simply reflecting the size of absolute peaks. The first subcluster captures how rainfall peaks dominate relative to the event average. *Peak—mean ratio* and *relative amplitude* form a tight group, both measuring peak intensity relative to the mean, while their association with the *coefficient of variation (CV)* reflects the shared emphasis on mean-based ratios. Although *CV* compares the mean to the standard deviation rather than the peak directly, the standard deviation rises sharply as peaks intensify, highlighting the contrast between intense and weaker timesteps. The second subcluster contains the moment-based descriptors *(C) kurtosis* and *(C) skewness*, which characterise the shape of the rainfall distribution. As negative kurtosis is rare in rainfall events (as this would involve lots of high intensity timesteps, accompanied by few low intensity), skewness and kurtosis generally co-vary. Both metrics tend to be high in events with rainfall concentrated in a few time steps, and low in more uniform events. Another subcluster tightly links the *% time in LIZ* with its complement, *% time in HIZ*, reflecting their inverse relationship, with *Lorenz asymmetry* joining nearby as it also quantifies imbalance in the rainfall distribution between wetter and drier periods. In a separate branch, *event-dry ratio*, *Gini coefficient*, and *% of rainfall in the HIZ* cluster together, collectively describing how rainfall volume concentrates into a few timesteps, with events scoring high on one metric tending to do so on the others as well.

The fifth cluster in Figure 9.a groups *temporal kurtosis*, *temporal standard deviation*, and the *TCI*, all of which quantify the temporal concentration of rainfall. Unlike magnitude-based concentration metrics, these measures only return high values when high-intensity timesteps are tightly grouped in time. Their very low correlation with other metrics indicates that this dimension of temporal loading is largely independent from others. This property is particularly relevant for applications such as flash-flood risk assessment and soil erosion modelling, where closely clustered high-intensity periods can greatly amplify impacts compared to if high intensity timesteps were spread across an event.

The remaining seven 'clusters' each contain only a single metric. In some cases this suggests they provide unique information about rainfall events. The *asymmetry of dependence* characterises how rainfall intensities evolve over time, identifying whether events increase and decrease in intensity at similar rates, a feature that may be relevant for understanding event evolution or storm generation. In Figure 9a, this metric is shown to split off very early from the other metrics, highlighting its unique position. The *event loading index* measures the deviation in temporal variability between an observed rainfall event and a version of that event but mirrored in time. This captures structural irregularities not represented by other metrics, although its conceptual similarity to asymmetry is reflected in its positioning between the asymmetry focused 1st and 2nd clusters.

Likewise, *intermittency* captures the frequency with which rainfall starts and stops during an event, a feature not reported by other metrics. In contrast, the isolated clustering of the fractional metrics, *fraction rainfall in Q2*, *Q3*, and *Q4*, more likely reflects their limited descriptive power when considered individually. Unless one quarter dominates the rainfall distribution, these metrics tend not to strongly distinguish between different event types or structures. Similarly, *mean intensity* does not intuitively convey any information about temporal loading. It is included in this analysis to test whether any other metrics are implicitly driven by the total event scale. This appears to not be the case, which is a useful conclusion. These results highlight the importance of selecting a diverse set of metrics when analysing rainfall temporal loading. While many metrics are highly interrelated, a subset offer complementary insights that may enhance interpretation or predictive performance when used together.

#### 5 Recommendations

A central contribution of this paper is the argument that analyses of rainfall temporal loading must begin with a clear definition of the aspect(s) most relevant to the research question or application. Rather than prescribing a single universal metric, we provide a conceptual framework to guide deliberate, defensible, and replicable metric selection. Based on the findings of the literature review and the data-driven clustering exercise, we recommend breaking event temporal loading into five components: (1) mass timing, (2) peak timing, (3) magnitude concentration, (4) temporal concentration, and (5) intermittency. In Table 3, for each component we summarise its meaning, list all associated metrics and indicate one suggested metric.

Our analysis shows that many metrics are sensitive to temporal resolution or DMC transformations. For peak timing, magnitude concentration, and intermittency, no metrics are fully robust, reflecting the inherent variability of peaks and the smoothing introduced by temporal aggregation. We highlight that this means that studies using these metrics across different resolutions should interpret comparisons with caution. For these components, we recommend use of metrics which have been more commonly applied in the past and which are conceptually simple. Temporal concentration and mass timing metrics are generally more robust. For these components, our recommended metrics also consider simplicity and prior usage in the literature, in addition to robustness. While we highlight the 'robust' metrics in Table 3, we note that Tables C1, C2 and C3 in Appendix C. provides full metric-specific summaries of robustness.

### 6 Summary and conclusions

545

This study provides a comprehensive review and empirical analysis of rainfall event temporal loading metrics, drawing on literature from multiple domains including flood modelling, soil erosion and sediment transport, pollution modelling, landslide prediction, and climate change impact assessment. We identify 52 metrics that have been used to describe the temporal distribution of rainfall within storms, and apply these to over 233,000 events observed by a large network of rain gauges. This represents the first large-scale, systematic evaluation of how such metrics behave across real-world rainfall events.

**Table 3.** Five recommended components of rainfall event temporal loading. Each component is accompanied by its definition and a list of all reviewed metrics which are best described by this component, with metrics found to be robust at different temporal resolutions and after DMC normalisation highlighted in red. A recommended metric for use studying each component is also provided.

| Category                        | Description                                                                                                                                                                                                                                       | All metrics                                                                                                                                                                                                                               | Recommendation                                            | on Reasons                                                                                                                                                                                                    |
|---------------------------------|---------------------------------------------------------------------------------------------------------------------------------------------------------------------------------------------------------------------------------------------------|-------------------------------------------------------------------------------------------------------------------------------------------------------------------------------------------------------------------------------------------|-----------------------------------------------------------|---------------------------------------------------------------------------------------------------------------------------------------------------------------------------------------------------------------|
| Mass timing                     | Describes when the majority of rainfall mass falls during the event. From the start (low values), through the middle, to the end (high values).                                                                                                   | 3rd / 4th / 5th with most,<br>3rd with D <sub>50</sub> , 3rd with<br>CoG, Centre of gravity,<br>D <sub>50</sub> , T25/75, m1, m3, m4,<br>m5, Event loading index,<br>(T) skewness, Asymmetry<br>of dependence, Frac. in Q1<br>/ 2 / 3 / 4 | 4th with most (categorical)  D <sub>50</sub> (continuous) | Of the robust metrics, these ones have been most commonly applied previously                                                                                                                                  |
| Peak timing                     | Describes when the peak intensity occurs during the event. From the start (low values), through the middle, to the end (high values).                                                                                                             | 3rd/4th/5th with peak, 3rd ppr, time-to-peak, peak-position-ratio, Skew <sub>p</sub>                                                                                                                                                      | Peak-position-<br>ratio                                   | No peak timing metrics are robust, so use in cross-study comparison with care. <i>Peak-position-ratio</i> is the most widely applied, easily understood option.                                               |
| Magnitude<br>concentra-<br>tion | Describes the strength of intense phases, contrasts between high- and low-intensity periods, or overall statistical inequality. Not affected by whether timesteps concentrating rainfall are close together in time, or where in time they occur. | Max intensity, I30, PCI, (C) skewness, (C) kurtosis, (C) std, CV, Mean intensity in HIZ, % time in LIZ/ HIZ, % Rain in HIZ, Gini Coefficient, Lorenz Asymmetry Coefficient, NRMSE <sub>p</sub> , Peak-mean ratio, Relative amplitude, m2  | Gini coefficient                                          | Gini coefficient is the only metric robust to both temporal resolution and DMC transformation.                                                                                                                |
| Temporal<br>concentra-<br>tion  | Describes how narrowly rainfall is clustered around a point in time, regardless of when in time that cluster appears.                                                                                                                             | TCI, T (kurtosis), T (std)                                                                                                                                                                                                                | (T) std                                                   | All metrics are robust. While <i>TCI</i> has been used previously in one study, we recommend that <i>T</i> ( <i>std</i> ) is a simpler metric to measure the same property.                                   |
| Intermittency                   | Describes how intermittent rainfall is within the event.                                                                                                                                                                                          | Intermittency fraction, Event dry ratio                                                                                                                                                                                                   | Event dry ratio                                           | Neither are robust across resolutions and so should be used in cross-study comparison with care. Both metrics have only been applied once previously, but the <i>event-dry ratio</i> is conceptually simpler. |

We begin by organising the metrics with a typology that groups metrics into three broad categories: (i) classification metrics, which assign events to discrete categories; (ii) summary statistics, which generate continuous descriptors of temporal loading; and (iii) intermittency metrics, which quantify the frequency and spacing of dry intervals within an event. This typology is constructed based primarily on metric form (e.g. categorical versus continuous) and purpose. We then perform a data-driven hierarchical cluster analysis that reveals further groups of metrics that transcend these categories. Notably, two well-defined groups of metrics are identified, including one capturing the timing of peak intensity and another describing the timing of rainfall mass distribution. Each includes both summary statistics and classification metrics, suggesting that despite differences in format, these metrics capture similar underlying aspects of temporal loading and tend to rank events similarly, indicating a degree of interchangeability for comparative purposes. However, we note that this does not necessarily imply that metric choice within these clusters will not materially alter overall findings. For instance, an analysis based on the 3rd with most rainfall may conclude differently on whether a geographical area generally has a tendency towards front-loaded events, than if  $D_{50}$  is the chosen metric. Exploring how variations in absolute metric values within correlated groups may alter substantive conclusions was beyond the scope of this study, but we highlight it as a valuable focus for future research.

The cluster analysis also reveals several looser groupings and a number of isolated metrics. This reflects a broader issue that many metrics used under the banner of 'temporal loading' in fact represent quite different concepts. This can lead to metrics being borrowed across domains without clear understanding of whether the metric is actually appropriate for the intended application. To address this, we propose a conceptual framework that distils our findings into five aspects of temporal loading: (1) mass timing, (2) peak timing, (3) magnitude concentration, (4) temporal concentration, and (5) intermittency. The choice of these five aspects is motivated by a combination of statistical evidence, existing practice, and conceptual need. Clarifying and standardising these concepts represents a critical step toward improving comparability and interpretability across studies.

We note that comparing metrics across different data types, particularly continuous summary statistics and discrete categories, introduces interpretive limitations. When calculating sMAPE scores and plotting metric distributions we opted to not transform or scale metrics. This was to preserve the physical interpretability of each metric, allowing percentage errors to reflect changes in real units (e.g., minutes or intensity proportions) rather than abstracted or standardised values. However, although sMAPE expresses the average percentage difference between two representations (e.g., at different resolutions), the meaning of a given percentage depends on the nature and scale of the metric. For instance, a 70% sMAPE for a bounded percentile-based timing measure like D<sub>50</sub> does not carry the same meaning as a 70% sMAPE for a scale-dependent metric such as standard deviation. This presents a trade-off, where preserving metric-specific meaning improves interpretability within each metric, but complicates comparative analysis across them. We recognise that we are operating within these constraints, but by interpreting sMAPE scores in conjunction with visual plots, rank correlations, and physical understanding of the metric's behaviour, we still make valuable assessments of metric behaviour at different resolutions and processing options.

Our empirical analysis of whether metrics report consistent results when applied to rainfall events at different temporal resolutions and when converted to normalised, dimensionless representations goes some way towards addressing the broader lack of methodological transparency in event temporal loading research. We show that some metrics, especially those based on the peak timing or magnitude, are highly sensitive to temporal resolution, whereas metrics capturing the overall precipitation

605

distribution tend to be more robust. Similar patterns apply for double normalised events, where the reduced number of time steps constrains accurate representation of peaks. Additionally, the de-dimensionalisation process leads to discretisation of some metric values and renders those based on absolute values uninterpretable. While these findings provide practical guidance for selecting appropriate metrics in contexts with limited or pre-processed data, a number of methodological uncertainties remain. In particular, it is often unclear whether metrics should be applied to the full rainfall event or to only the most intense burst.
 Additionally, the treatment of events with multiple peaks, or with no clear peak, remains insufficiently explored. Many metrics assume events have a single dominant peak, but the implications of including or excluding multi-peaked or uniform events in analyses are not well understood. Some research, e.g., Nguyen and Chen (2022), attempts to separate uniform rainfall events, using the Schutz Index, and to treat these differently. We recommend future research investigate this further, and consider filtering approaches based on peak structure, including the potential exclusion of events with multiple or indistinct peaks, to
 ensure appropriate and meaningful application of temporal loading metrics.

Finally, it is important to reflect on the representativeness of the rainfall data used in this study. Our analysis draws on the high-quality, dense network of gauges across Denmark, providing over 233,000 well-observed rainfall events. This scale of dataset allows for robust evaluation of metric behaviour under real storm conditions. However, Denmark's temperate maritime climate is not fully representative of the range of rainfall regimes globally. In particular, strongly convective extremes typical of tropical and subtropical climates are less frequent, and storm dynamics may differ from those in arid or mountainous regions. As such, while our findings on the behaviour, sensitivity, and clustering of metrics are expected to generalise across contexts, the specific distribution of rainfall events and the relative prevalence of different temporal loading structures may not. We therefore recommend that future work apply similar analyses in a wider set of climatic settings, both to test the robustness of our methodological conclusions and to build a more globally representative basis for metric selection.

10 Code availability. Code is available at https://github.com/masher92/MetricEvaluation/

## Appendix A: Boolean search query

Scopus supports proximity operators (e.g., W/n requires two words to appear within n words of each other; Pre/n requires one word to precede the other within n words), a feature not available in the other databases. As such, for Scopus the search term was: [(("rainfall event" OR "precipitation event" OR "rain storm" OR "rainstorm" OR "storm event" OR (( 'precipitation" OR "rain" OR "rainfall" OR "storm") PRE/3 "event")) OR (( "precipitation varia\*" OR "rainfall varia\*") W/1 "intra-event")) W/5 ("rainfall profile" OR "temporal pattern" OR "temporal distribution" OR "event loading" OR "temporal profile" OR "temporal loading" OR "intensity profile" OR (( "precipitation pattern" OR "rainfall pattern") W/3 "temporal"))] and for Web of Science and Google Scholar, it was: ("rainfall event" OR "precipitation event" OR "rain storm" OR "rainstorm" OR "storm event" OR ("precipitation" OR "rainfall" OR "rainfall" OR "storm") AND "event" OR ( "precipitation varia\*" OR "rainfall varia\*") AND

"intra-event") AND ("rainfall profile" OR "temporal pattern" OR "temporal distribution" OR "event loading" OR "temporal profile" OR "temporal loading" OR "intensity profile" OR ("precipitation pattern" OR "rainfall pattern") AND "temporal")].

# **Appendix B: Selected papers**

Table B1. Summary of reviewed papers

| Authors                              | Metric type                      | Metric                                    | Study purpose                                                | Rainfall  | Processing |
|--------------------------------------|----------------------------------|-------------------------------------------|--------------------------------------------------------------|-----------|------------|
| Amin et al. (2000)                   | Classification                   | 4th with most                             | Characterising typical profiles in an area                   | Real      | DMC        |
| Aquino et al. (2013)                 | Classification                   | 3rd with peak                             | Impact of storm temporal pattern on Real soil erosion        |           | Raw        |
| Asher et al. (2025)                  | Classification                   | 5th with most                             | Impact of storm temporal pattern on flood risk               | Synthetic | DMC        |
| Azli and Rao (2010)                  | Classification                   | 4th with most                             | Characterising typical profiles in an area                   | Real      | DMC        |
| Back and Rodrigues (2021)            | Classification                   | 4th with peak                             | Characterising typical profiles in an area                   | Real      | Unclear    |
| Bezak et al. (2018)                  | Classification                   | 4th with most                             | Impact of storm temporal pattern on hydraulic model outcomes | Synthetic | DMC        |
| Brommer et al. (2013)                | Summary stats                    | mean, (C) std, (C) skewness, (C) kurtosis | Characterising typical profiles in an area                   | Real      | Raw        |
| Cai et al. (2024)                    | Summary stats                    | Peak-position-<br>ratio, PCI              | Impact of storm temporal pattern on peak runoff generation   | Synthetic | Raw        |
| Chen et al. (2015)                   | Classification                   | 4th with most                             | Characterising typical profiles in an area                   | Real      | Unclear    |
| de Assunção Montenegro et al. (2018) | Classification                   | 3rd with peak                             | Impact of storm temporal pattern on soil moisture patterns   | Real      | Raw        |
| de Andrade et al. (2020)             | Classification,<br>Summary stats | 130, 3rd with peak                        | <u> </u>                                                     |           | Raw        |
| Dolšak et al. (2016)                 | Classification                   | 4th with most, BSC                        | C Characterising typical profiles in an Real area            |           | DMC        |
| Dunkerley (2021b)                    | Intermittency                    | Intermittency frac-                       | Developing methods for describing temporal profile           | Real      | Raw        |
| Fan et al. (2020)                    | Summary stats                    | (C) skewness, (C) kurtosis                | Impact of storm temporal pattern on landslides               | Synthetic | Raw        |

| Authors                                       | Metric type                  | Metric                                            | Study purpose                                                            | Rainfall  | Processing |
|-----------------------------------------------|------------------------------|---------------------------------------------------|--------------------------------------------------------------------------|-----------|------------|
| Fatone et al. (2021)                          | Classification               | 3rd with peak                                     | Impact of storm temporal pattern on hydrograph outflow                   | Real      | DMC        |
| Fu et al. (2021)                              | Summary stats                | Coefficient peak rainfall intensity, 3rd with ppr | Impact of typhoon temporal pattern on pollution generation               | Real      | Raw        |
| Gao et al. (2024)                             | Summary stats                | Relative amplitude                                | Impact of storm temporal pattern on soil erosion                         | Real      | Raw        |
| García-Bartual and Andrés-<br>Doménech (2017) | Summary stats                | (C) kurtosis, (C) standard deviation              | Characterising typical profiles in an area                               | Real      | Raw        |
| Garcia-Guzman and Aranda-<br>Oliver (1993)    | Classification               | 4th with most                                     | Characterising typical profiles in an area                               | Real      | DMC        |
| Ghanghas et al. (2024)                        | Summary stats                | Event Loading Index                               | Impact of climate on temporal pattern                                    | Real      | Raw        |
| He et al. (2022)                              | Summary stats                | PCI                                               | Characterising typical profiles in an area                               | Synthetic | Raw        |
| He et al. (2024)                              | Summary stats                | (C) skewness, (C)<br>kurtosis                     | Impact of storm temporal pattern on slope instability                    | Synthetic | Raw        |
| Horner and Jens (1942)                        | Classification               | Third with peak                                   | Impact of storm temporal pattern on surface runoff                       | Real      | Unclear    |
| Hu et al. (2018)                              | Summary stats                | Gini coefficient,<br>Lorenz coefficient           | Changes to temporal inequality of daily precipitation over last 50 years | Synthetic | Raw        |
| Huff (1967)                                   | Classification               | 4th with most                                     | Characterising typical profiles in an area                               | Real      | DMC        |
| Jeon et al. (2017)                            | Classification               | 4th with most                                     | Impact of storm temporal pattern on water quality from storm runoff      | Synthetic | DMC        |
| Jun et al. (2021)                             | Summary stats Classification | 4th with most, time to peak                       | Characterising typical profiles in an area                               | Real      | Unclear    |
| Knighton and Walter (2016)                    | Summary stats                | T25   50   75, time<br>to peak                    | Stochastic event generation                                              | Real      | DMC        |
| Li et al. (2023)                              | Summary stats                | Peak-mean ratio, peak-position-ratio              | Evaluating satellite precipitation products                              | Real      | Raw        |
| Liang et al. (2023)                           | Classification               | 3rd with most                                     | Impact of storm temporal pattern on soil erosion                         | Synthetic | DMC        |

| Li et al. (2021)              | Classification | 4th with most          | Characterising typical profiles in an | Real      | Raw     |
|-------------------------------|----------------|------------------------|---------------------------------------|-----------|---------|
| L1 Ct al. (2021)              | Ciassification | +ın wun most           | area;                                 | Keai      | Naw     |
|                               |                |                        | Impact of storm temporal pattern on   |           |         |
|                               |                |                        | sewer overflows                       |           |         |
| Live et al. (2022)            | Cummany atata  | Dalatina amalituda     |                                       | Dool      | Daw     |
| Liu et al. (2022)             | Summary stats  | Relative amplitude,    | Impact of storm temporal pattern on   | Real      | Raw     |
|                               |                | % rain HIZ, % time     | runoff and soil loss                  |           |         |
|                               |                | in HIZ / LIZ           |                                       |           |         |
| Long et al. (2021)            | Summary stats  | TCI                    | Impact of climate on temporal (and    | Real      | Unclear |
|                               |                |                        | spatial) pattern                      |           |         |
| Loveridge et al. (2015)       | Classification | 3rd with D50           | Characterising typical profiles in an | Real      | DMC     |
|                               |                |                        | area                                  |           |         |
| Lu and Qin (2020)             | Classification | 4th with most          | Impact of storm temporal pattern on   | Synthetic | DMC     |
|                               |                |                        | urban drainage                        |           |         |
| Müller et al. (2017)          | Summary stats  | Asymmetry of de-       | Impact of storm temporal patterns     | Real      | Raw     |
|                               |                | pendence               | on sewer flow simulations             |           |         |
| Nel (2007)                    | Classification | 4th with peak          | Characterising typical profiles in an | Real      | DMC     |
|                               |                |                        | area;                                 |           |         |
|                               |                |                        | Impact of storm temporal pattern on   |           |         |
|                               |                |                        | runoff and soil erosion               |           |         |
| Oh et al. (2024)              | Summary stats  | $NRMSE_p$ and $Skew_p$ | Impact of storm temporal pattern on   | Real      | Unclear |
| ,                             | •              | r                      | peak flood discharge                  |           |         |
| Pinheiro et al. (2018)        | Classification | 3rd with peak          | Characterising typical profiles in an | Real      | DMC     |
| , ,                           |                |                        | area;                                 |           |         |
|                               |                |                        | Impact of storm temporal pattern on   |           |         |
|                               |                |                        | soil erosion                          |           |         |
| Pohle et al. (2018)           | Summary stats; | Event dry ratio,       | Stochastic event generation           | Synthetic | Raw     |
| ()                            | Intermittency  | Frac. in Q1 / Q2 /     |                                       |           |         |
|                               |                | Q3 / Q4                |                                       |           |         |
| Terranova and Iaquinta (2011) | Classification | BSC                    | Characterising typical profiles in an | Real      | DMC     |
| 1011ano (a ana 1aquina (2011) | Siassiffoution | 250                    | area                                  | 11041     | Direc   |
| Tian et al. (2017)            | Summary stats  | CV                     | Stochastic event generation           | Synthetic | Raw     |
| Todisco (2014)                | Classification | CV, Run number         | Developing methods for describing     | Real      | Raw     |
| 1041300 (2017)                | Ciassification | Cr, Run number         | temporal profile;                     | icai      | ixaw    |
|                               |                |                        |                                       |           |         |
|                               |                |                        | Impact of storm temporal pattern on   |           |         |
| m 1 (2005)                    | al in          |                        | soil erosion                          |           | D       |
| Tsubo et al. (2005)           | Classification | 4th with peak          | Characterising typical profiles in an | Real      | DMC     |
|                               |                |                        | area                                  |           |         |

| Serna and Taipe (2019)           | Classification | 3rd with peak                                  | Characterising typical profiles in an area                                                           | Real | DMC     |
|----------------------------------|----------------|------------------------------------------------|------------------------------------------------------------------------------------------------------|------|---------|
| Schiff (1943)                    | Classification | 3rd with peak                                  | Impact of storm temporal pattern on infiltration and runoff                                          | Real | DMC     |
| Varga et al. (2009)              | Classification | 3rd with CoG                                   | Developing methods for describing temporal profile                                                   | Real | DMC     |
| Vantas et al. (2019)             | Classification | 4th with most                                  | Developing methods for describing temporal profile                                                   | Real | Raw     |
| Villalobos Herrera et al. (2023) | Classification | 5th with most                                  | Characterising typical profiles in an area                                                           | Real | DMC     |
| Visser et al. (2023)             | Summary stats  | $D_{50}$                                       | Impact of climate on temporal pattern                                                                | Real | DMC     |
| Wang et al. (2016)               | Classification | 3rd with peak                                  | Impact of storm temporal pattern on soil erosive loss                                                | Real | DMC     |
| Wang et al. (2018)               | Classification | 3rd with ppr                                   | Impact of storm temporal pattern on soil water transport                                             | Real | DMC     |
| Yang et al. (2015)               | Summary stats  | Coefficient of vari-<br>ability                | Assessing performance of data assimilation on events with different temporal loading characteristics | Real | Raw     |
| Yang et al. (2022)               | Summary stats  | Peak-position-ratio                            | Characterising typical profiles in an area  Regional patterns in temporal profile                    | Real | Raw     |
| Yang et al. (2024)               | Classification | 5th with peak                                  | Impact of storm temporal pattern on flash flooding                                                   | Real | Unclear |
| Yin et al. (2016)                | Classification | 4th with peak                                  | Characterising typical profiles in an area                                                           | Real | DMC     |
| Wartalska et al. (2020)          | Summary stats  | m1, m2, m3, m4,<br>m5, peak-position-<br>ratio | Characterising typical profiles in an area                                                           | Real | DMC     |
| Wartalska and Kotowski (2020)    | Summary stats  | $m4, n_i$                                      | Characterising typical profiles in an area; Impact of future climate on typical profiles.            | Real | DMC     |
| Williams-Sether (2004)           | Classification | 4th with most                                  | Characterising typical profiles in an area                                                           | Real | DMC     |

# **Appendix C: Selected metrics**

Table C1. Summary of reviewed metrics (Part 1: Classification)

| Metric                | Meaning                                        | Focus       |                         | Robust to: |
|-----------------------|------------------------------------------------|-------------|-------------------------|------------|
|                       |                                                |             | Temp res DMC transforma |            |
| 3rd/4th/5th with peak | Fraction of event with peak rainfall           | Peak timing | ×                       | ×          |
| 3rd/4th/5th with most | Fraction of event with most rainfall           | Asymmetry   | $\checkmark$            | ✓          |
| $3rd$ with $D_{50}$   | Third in which 50% of rainfall is reached      | Asymmetry   | $\checkmark$            | ✓          |
| 3rd with CoG          | Third containing centre of gravity of rainfall | Asymmetry   | ✓                       | ✓          |

Table C2. Summary of reviewed metrics (Part 2: Summary statistics)

| Metric                                | Meaning                                                 | Focus        |          | Robust to:         |
|---------------------------------------|---------------------------------------------------------|--------------|----------|--------------------|
|                                       |                                                         |              | Temp res | DMC transformation |
|                                       | Peakiness indicators                                    |              |          |                    |
| Max intensity                         | Maximum rainfall intensity                              | Peakiness    | ×        | ×                  |
| Mean intensity                        | Mean rainfall intensity                                 | Peakiness    | ✓        | ×                  |
| <i>I30</i>                            | Max rainfall depth in any 30-minute period              | Peakiness    | ✓        | ×                  |
| Classical (C) std                     | Standard deviation: variation of rainfall intensity     | Peakiness    | ✓        | ×                  |
|                                       | values around mean intensity                            |              |          |                    |
| Cv                                    | Coefficient of variation (std/mean): standardised       | Peakiness    | ✓        | ×                  |
|                                       | variation of intensity values around mean intensity     |              |          |                    |
| Classical (C) skewness                | Asymmetry of intensity value distribution. + skew-      | Peakiness    | ×        | ×                  |
|                                       | ness indicates most values small, some big, - skew-     |              |          |                    |
|                                       | ness indicates most values big, some small.             |              |          |                    |
| Classical (C) kurtosis                | Tailedness of intensity value distribution. + kur-      | Peakiness    | ×        | ×                  |
|                                       | tosis indicates sharp, intense bursts concentrated      |              |          |                    |
|                                       | in short periods, - kurtosis reflects rainfall spread   |              |          |                    |
|                                       | more evenly over time without extreme peaks.            |              |          |                    |
| Peak-mean ratio / Rainfall            | Peak-to-mean ratio: maximum intensity divided by        | Peakiness    | ×        | ×                  |
| intensity irregularity $(n_i)$        | mean intensity                                          |              |          |                    |
| Relative amplitude (R <sub>am</sub> ) | Range divided by mean                                   | Peakiness    | ×        | ×                  |
| m2                                    | % rainfall in highest-intensity time step / rainfall in | Peakiness    | ×        | ×                  |
|                                       | whole event                                             |              |          |                    |
| Mean intensity in HIZ                 | Average intensity during above-average intensity        | Peakiness    | ✓        | ×                  |
|                                       | periods                                                 |              |          |                    |
|                                       | Asymmetry indicator                                     | rs           |          |                    |
| Time to peak                          | Time to peak in minutes                                 | Peak timing  | ×        | ×                  |
| Peak-position-ratio /                 | Time to peak divided by duration                        | Peak timing  | ×        | ×                  |
| coefficient-peak-rainfall-            |                                                         |              |          |                    |
| intensity)                            |                                                         |              |          |                    |
| $Skew_p$                              | Value from -0.25 to 0.25 measuring relative position    | Peak timing  | ×        | ×                  |
|                                       | of peak within rainfall duration indicates front-       |              |          |                    |
|                                       | loading, + indicates back-loading.                      |              |          |                    |
| T25 / T50 and D <sub>50</sub> / T75   | Percent of event passed when X% of rainfall hap-        | Asymmetry of | ✓        | ✓                  |
|                                       | pened                                                   | rainfall     |          |                    |
| m3 / m4 / m5                          | % rainfall in first 33% / 30% / 50% of event duration   | Asymmetry of | ✓        | ✓                  |
|                                       |                                                         | rainfall     |          |                    |

| Metric                      | Meaning                                                   | Focus          | Robust to:   |                    |  |
|-----------------------------|-----------------------------------------------------------|----------------|--------------|--------------------|--|
|                             |                                                           |                | Temp res     | DMC transformation |  |
| m1                          | Relative volume of rainfall before versus after the       | Asymmetry of   | ×            | ×                  |  |
|                             | peak                                                      | rainfall       |              |                    |  |
| Centre of gravity (CoG)     | Timing of rainfall mass centre, snapped to nearest        | Asymmetry of   | ✓            | <b>√</b>           |  |
|                             | time step                                                 | rainfall       |              |                    |  |
| Centre of gravity (interpo- | Fractional timing of rainfall mass centre across          | Asymmetry of   | ✓            | ✓                  |  |
| lated) (CoG interp.)        | event duration                                            | rainfall       |              |                    |  |
| Event loading index         | Comparison of temporal variability (captured by           | Asymmetry of   | ×            | ×                  |  |
|                             | the standardised temporal heterogeneity, STH) of a        | rainfall       |              |                    |  |
|                             | storm against that of a symmetrised version con-          |                |              |                    |  |
|                             | structed by mirroring the rising limb around the          |                |              |                    |  |
|                             | peak.                                                     |                |              |                    |  |
| Asymmetry of dependence     | Describes the degree of reversibility in a timeseries     | Asymmetry of   | ×            | ×                  |  |
|                             | using ranked-based statistics (copulas)                   | rainfall       |              |                    |  |
| Temporal (T) skewness       | Skewness of rainfall distribution considered over         | Asymmetry of   | $\checkmark$ | $\checkmark$       |  |
|                             | $time-more\ front-loaded, +more\ back-loaded$             | rainfall       |              |                    |  |
|                             | Concentration indicate                                    | ors            |              |                    |  |
| Precipitation Concentra-    | Compares sum of squared intensities to square of          | Magnitude con- | ×            | ×                  |  |
| tion Index (PCI)            | total rainfall. Higher values indicate more rainfall      | centration     |              |                    |  |
|                             | concentrated in fewer steps.                              |                |              |                    |  |
| Temporal Concentration      | Measures how strongly rain is clustered in time           | Temporal con-  | $\checkmark$ | $\checkmark$       |  |
| Index (TCI)                 | around each time-step, selects value for time step        | centration     |              |                    |  |
|                             | around which rainfall is most tightly clustered.          |                |              |                    |  |
|                             | A higher TCI means a more tightly concentrated            |                |              |                    |  |
|                             | "core" of high rain intensity                             |                |              |                    |  |
| $NRMSE_p$                   | Quantifies concentration of rainfall near the peak        | Temporal Con-  | ×            | ×                  |  |
|                             | (RMSE from peak, normalised by rainfall total). 0         | centration     |              |                    |  |
|                             | = uniform distribution, $1$ = all rainfall in single time |                |              |                    |  |
|                             | step.                                                     |                |              |                    |  |
| Gini coefficient            | Measures inequality of rainfall distribution (higher      | Magnitude con- | $\checkmark$ | $\checkmark$       |  |
|                             | values indicate greater concentration of rainfall in      | centration     |              |                    |  |
|                             | fewer intervals)                                          |                |              |                    |  |
| Lorenz asymmetry coeffi-    | Quantifies skewness of rainfall intensity distribu-       | Magnitude con- | ×            | ×                  |  |
| cient                       | tion relative to the mean, distinguishing whether         | centration     |              |                    |  |
|                             | the concentration arises from a few extreme high-         |                |              |                    |  |
|                             | intensity bursts or from many moderately intense          |                |              |                    |  |
|                             | intervals, even if overall inequality is similar.         |                |              |                    |  |

| Metric                | Meaning                                               | Focus          |          | Robust to:         |
|-----------------------|-------------------------------------------------------|----------------|----------|--------------------|
|                       |                                                       |                | Temp res | DMC transformation |
| % time in LIZ / HIZ   | Time spent below / above mean intensity. Indicates    | Magnitude con- | ×        | ×                  |
|                       | how long the storm stays intense for.                 | centration     |          |                    |
| % rain in HIZ         | Proportion of rainfall in above-average intensity pe- | Magnitude con- | ×        | ×                  |
|                       | riods. More rainfall in HIZ indicates a more concen-  | centration     |          |                    |
|                       | trated event.                                         |                |          |                    |
| Frac. in Q1/Q2/Q3/Q4  | % total rainfall in each quarter                      | Magnitude con- | ×        | ×                  |
|                       |                                                       | centration     |          |                    |
| Temporal (T) kurtosis | Whether rainfall is very concentrated in time around  | Temporal con-  | ✓        | $\checkmark$       |
|                       | the centre of gravity, or spread out in the tails     | centration     |          |                    |
| Temporal (T) std      | How spread out rainfall is in time around centre of   | Temporal con-  | ✓        | ✓                  |
|                       | gravity, weighted by heaviness of rain                | centration     |          |                    |

 Table C3. Summary of reviewed metrics (Part 3: Intermittency metrics)

| Metric                 | Meaning                                       | Focus         |          | Robust to:         |
|------------------------|-----------------------------------------------|---------------|----------|--------------------|
|                        |                                               |               | Temp res | DMC transformation |
| Event dry ratio        | Proportion of event duration spent at 0 mm/hr | Intermittency | ×        | ×                  |
| Intermittency fraction | Frequency of wet/dry transitions in the event | Intermittency | ×        | ×                  |

### Appendix D: Silhoutte score

Figure D1. Silhouette scores for different numbers of clusters

Author contributions. Conceptualisation and methodology was by MA and JP, Funding acquisition and supervision by MT and CB. Data curation was by MA, JP and RH. Formal analysis was by MA and RH. Investigation and visualisation was by MA. The original manuscript was written by MA, with review and editing by JP, MT, CB and SB.

Competing interests. The authors declare that they have no conflict of interest.

Acknowledgements. We would like to thank the Water Pollution Committee of The Society of Danish Engineers for allowing us to use their data for this study.

We also extend thanks to Lawrence Jackson for his guidance on the statistical analysis of metrics performed in this study.

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
