# Peer review of "Critical assessment of metrics and methods used to quantify temporal loading of rainfall events"

_EGUsphere, 2025_

## Referee Comment (RC1)

**Review Comment hess-2022-94**

**Title: Critical assessment of metrics and methods used to quantify temporal loading of rainfall events**

In this paper a large set of (52) metrics are investigated, that describe temporal loading (i.e. temporal evolution) of rainfall events. This first part constitutes a rather elaborate literature review that concludes with a summary table listing the various metrics and in how many studies the authors found them to be used. Then, these metrics are calculated for a large number (>100,000) rainfall events recorded at Danish rain gauges between 1975 and 2025. The metrics are evaluated with respect to their degree of overlap (redundancy) and robustness to changes in temporal resolution (aggregation) and choices made during data processing. The study aims to answer four research questions: RQ1: What key properties of rainfall event temporal loading are commonly measured, and why? RQ2: How sensitive are these metrics to the temporal resolution of the rainfall data? RQ3: How does de-dimensionalisation of rainfall events affect metric values? RQ4: Which metrics are strongly correlated, suggesting they may be redundant or are suitable for use in cross-comparison studies.

The study represents a diligent amount of work, but what does not become clear after reading through the results and conclusions, is a justification for why this work is needed. A summary and comparison of the many metrics is surely informative, but the question remains what new insights we can gain from it. What can we not do now that this evaluation of metrics will enable us to do? One of the reasons mentioned for conducting the study is the need to incorporate temporal loading characteristics in the development of design storms for flood modeling and flood risk assessment. Representing temporal loading of storms in a statistically representative way is indeed not straightforward and new ideas or insights could be really valuable for the field. It seems like a missed opportunity that a study of such a large number of rainfall events restricts itself to just descriptive metrics and does not provide any statistical analysis that could feed into, for instance, recommendations for calculation of return periods in flood risk analysis and development of design storms.

In its current version, the analyses seems more suitable for submission in the form of a technical note. This would require drastically shortening the content, some suggestions are provided in the following.

**Detailed comments:**

- 1. Introduction: Several statements are made here that would benefit from a cited reference. A couple of examples
  - P2, L 39: "rainfall event temporal loading is often oversimplified or overlooked in impact modeling applications." Not sure that this still represents current practice?
  - P2, L 44: "symmetrical, centrally peaked intensity profiles are commonly used in flood modeling (..)". Many other approaches are used for flood modeling these days. Please place statement into context
  - P3, L58: "The relevance of each aspect varies by (...) and can result in misplaced emphasis and misinterpretation". Please provide a reference for this statement?
- 2. Literature review: this review covers 7.5 pages summarizing metrics found in the literature that are then summarized in a nice overview table. The information density of this section is quite low (it's very wordy), and could easily be summarized in just the table with short descriptions of the metrics (Metric name | Metric description | References of studies where metric was used).

**3. Methods:**

L 275: A peculiar statement is made here that requires better justification: "For temporal loading, which is interested in what happens around the peak, the way in which the edges of the event are defined is particularly important, but in this research we do not investigate this further."
If this aspect is so important, and central to the subject of study, should it not be particularly addressed?

- L294: "Rainfall temporal loading metrics are implemented in Python based on the definitions provided in the original publications"

It strikes me as odd that in a study that focuses on the calculation of metrics, no equations are provided of how metrics are calculated.

**4. Results and discussion**

- 4.1 Sensitivity of temporal aggregation:
  - the findings in this subsection on the effects of temporal aggregation are rather obvious, namely that values of peak intensities are particularly sensitive to temporal aggregation. It doesn't really seem worth analyzing and reporting?
- 4.3 Metric redundancy: the same comment applies here: is it really a new insight that metrics related to time of the peak are related, and similar for mass distribution etc.

**5. Recommendations**

"A central contribution of this paper is the argument that analyses of rainfall temporal loading must begin with a clear definition of the aspects most relevant to the research question or application."

This seems like a rather generic and common sense argument. Which comes back to the earlier comment that the new insight provided by this study is not very clear.

**6. Summary and conclusions**

This section should preferably restrics itself to presenting the conclusions. A short summary is already provided in the abstract of the paper.

---

## Author Comment (AC1)

**Appendix: Response to Reviewers**

**Reviewer 1 - Marie-Claire ten Veldhuis**

In this paper a large set of (52) metrics are investigated, that describe temporal loading (i.e. temporal evolution) of rainfall events. This first part constitutes a rather elaborate literature review that concludes with a summary table listing the various metrics and in how many studies the authors found them to be used. Then, these metrics are calculated for a large number (>100,000) rainfall events recorded at Danish rain gauges between 1975 and 2025. The metrics are evaluated with respect to their degree of overlap (redundancy) and robustness to changes in temporal resolution (aggregation) and choices made during data processing. The study aims to answer four research questions: RQ1: What key properties of rainfall event temporal loading are commonly measured, and why? RQ2: How sensitive are these metrics to the temporal resolution of the rainfall data? RQ3: How does de-dimensionalisation of rainfall events affect metric values? RQ4: Which metrics are strongly correlated, suggesting they may be redundant or are suitable for use in cross-comparison studies.

The study represents a diligent amount of work, but what does not become clear after reading through the results and conclusions, is a justification for why this work is needed. A summary and comparison of the many metrics is surely informative, but the question remains what new insights we can gain from it. What can we not do now that this evaluation of metrics will enable us to do? One of the reasons mentioned for conducting the study is the need to incorporate temporal loading characteristics in the development of design storms for flood modeling and flood risk assessment. Representing temporal loading of storms in a statistically representative way is indeed not straightforward and new ideas or insights could be really valuable for the field. It seems like a missed opportunity that a study of such a large number of rainfall events restricts itself to just descriptive metrics and does not provide any statistical analysis that could feed into, for instance, recommendations for calculation of return periods in flood risk analysis and development of design storms.

In its current version, the analyses seems more suitable for submission in the form of a technical note. This would require drastically shortening the content, some suggestions are provided in the following.

**General reply - Review 1**

> Thank you for your comments and review. We appreciate the recognition of the scale of the work, and acknowledge your critique that the original manuscript did not sufficiently make the case for why such a comprehensive evaluation of rainfall temporal loading metrics is needed, nor what new insight or capability it provides. In response to this overarching point, conveyed in Comments 1.5-1.7, we have revised the manuscript to make clearer that the main purpose of this work was never to propose new metrics, or to offer specific guidance on design storm development, rather to advance development of a conceptual and empirical framework that supports consistent interpretation and comparison of results across studies, and more deliberate selection of metrics in future analyses.

The study is intentionally structured around two linked components. The literature review establishes how rainfall temporal loading is conceptualised and quantified across different research communities, highlighting substantial variation in terminology and metric usage. This novel synthesis is a key scientific contribution of this work, and provides the necessary context for the large-scale empirical evaluation of 52 metrics across more than 200,000 rainfall events. This empirical analysis quantifies robustness to aggregation and calculation on dimensionless mass curves, and studies the relationships between metrics. Without the literature-based context, the empirical analysis would be difficult to interpret and its relevance for cross-study comparison would be limited.

We acknowledge the reviewer's suggestion that the analysis might be more suitable for a technical note, but we believe that such a format would require removal of the synthesis that underpins the empirical contribution. In particular, the ability to interpret metric redundancy and robustness in a generalisable way relies on first establishing how and why these metrics are used across the literature. For this reason, we maintain that the Research Article format remains appropriate, while recognising the need for greater concision and clarity.

To address these concerns, we have substantially shortened the literature review from 7.5 to 5.5 pages (approximately a 30% reduction), with the review methodology, Figure 3, and Table 1 moved to an appendix. The remaining text has been edited to reduce verbosity and improve focus. In addition, the Introduction and Discussion have been revised to more clearly motivate the need for a coherent metric categorisation and to explicitly state the intended contribution of the work. In particular, we now state at the end of the Introduction that the paper develops a conceptual categorisation of temporal loading aspects, based on the four research questions, to support cross-study interpretation and meaningful metric selection in future research.

We hope that these revisions clarify both the motivation for the study and the nature of its contribution: not the introduction of new metrics or direct design-storm prescriptions, but a framework for understanding what different rainfall temporal loading metrics measure, when they are comparable, and when they are not.

**Question 1.1.** *Introduction: Several statements are made here that would benefit from a cited reference. A couple of examples: - P2, L 39: "rainfall event temporal loading is often oversimplified or overlooked in impact modeling applications." Not sure that this still represents current practice? - P2, L 44: "symmetrical, centrally peaked intensity profiles are commonly used in flood modeling (..)". Many other approaches are used for flood modeling these days. Please place statement into context - P3, L58: "The relevance of each aspect varies by (...) and can result in misplaced emphasis and misinterpretation". Please provide a reference for this statement?*

**Response 1.1.**

We thank the reviewer for these comments and agree that several statements in the Introduction could benefit from clearer contextualisation and supporting references.

Regarding the statement that rainfall event temporal loading is often oversimplified or overlooked in impact modelling applications, our intention was not to suggest that this reflects all current practice, but rather that simplified temporal representations remain widely used in many design and applied modelling contexts. We have revised the text to make this scope clearer and to explicitly link this statement to commonly used design storm approaches, such as the FEH and Chicago profiles, which are discussed immediately thereafter.

Likewise, with respect to the comment that many other approaches are now used in flood modelling, we agree. Continuous simulation and event-based modelling using observed or stochastic rainfall are increasingly applied. We have amended the text to acknowledge these approaches, while noting that design storm methods remain prevalent in practice due to their simplicity and regulatory acceptance.

Finally, regarding the statement that the relevance of different aspects of temporal loading varies by application and can lead to misplaced emphasis, we agree that this is a general observation rather than a single result attributable to one study. We have clarified this point by briefly contrasting possible sensitivities across different application domains (e.g. flood modelling versus soil erosion).

**Question 1.2.** *Literature review: this review covers 7.5 pages summarizing metrics found in the literature that are then summarized in a nice overview table. The information density of this section is quite low (it's very wordy), and could easily be summarized in just the table with short descriptions of the metrics (Metric name| Metric description | References of studies where metric was used).*

**Response 1.2.**

We have carefully revised the manuscript to address this concern. The literature review has been substantially shortened (by approximately 30% / 2 pages), with the literature review methods and Figure 3 and Table 1 relocated to an appendix. The remaining sections have been edited throughout to reduce verbosity and improve readability. Additionally, we note that Tables B1 and C1-C3 in the appendices already summarise the information suggested as useful by the reviewer.

**Question 1.3.** *Methods: - L 275: A peculiar statement is made here that requires better justification: "For temporal loading, which is interested in what happens around the peak, the way in which the edges of the event are defined is particularly important, but in this research we do not investigate this further." If this aspect is so important, and central to the subject of study, should it not be particularly addressed?*

**Response 1.3.**

We thank the reviewer for highlighting this point and agree that the original wording could be interpreted as inconsistent. We agree with the reviewer that event definition is a critically important factor in shaping rainfall event profiles and can substantially influence the timing, position, and relative magnitude of peak intensity, as well as derived temporal loading metrics. The decision not to explicitly explore the sensitivity of temporal loading metrics to event definition was therefore not based on a lack of importance, but on considerations of scope. The influence of event boundary definition has been previously examined in the literature, with multiple studies demonstrating its effects on rainfall intensity, duration, depth, and inferred event characteristics. These studies collectively show that event definition alone can materially alter conclusions drawn from rainfall analyses.

In contrast, the focus of this paper is on a set of methodological choices that have received comparatively little systematic attention, namely the definition of the analytical objective (i.e. which aspect of temporal loading is of interest), the selection of metrics used to represent that objective, and the sensitivity of those metrics to rainfall processing choices such as temporal aggregation and normalisation. Expanding the analysis to include multiple alternative event definitions would have substantially increased the dimensionality of the problem and risked obscuring the specific effects this study seeks to isolate.

To address the reviewer's concern, we have expanded the discussion in the manuscript so the paragraph now reads as follows:

**To ensure event independence, we extract events using a minimum inter-event time (MIT) threshold (Restrepo-Posada and Eagleson, 1982; Molina-Sanchis et al., 2016). An 'event' thus constitutes any rainfall separated by at least 11 hours of rain-free conditions, following practice in several Danish hydrological studies (Gregersen et al., 2013; Thomassen et al., 2023). This approach ensures that each event begins and ends with non-zero rainfall. The choice of MIT has been shown to play an important role in determining both the number and properties of rainfall events identified (Dunkerley, 2008). In this study, event definition is treated as a fixed preprocessing choice rather than a variable of investigation, reflecting a deliberate scoping decision. While the delineation of event boundaries can influence the timing and relative prominence of peak intensity, and hence derived temporal loading metrics, its effects have been examined in several previous studies, e.g. Dunkerley (2008, 2010, 2015); Wang et al. (2019); Freitas et al. (2020); Molina-Sanchis et al. (2016); Haile et al. (2011); Medina-Cobo et al. (2016); Meier et al. (2016). In contrast, this study focuses on methodological choices that have received less systematic attention, namely the selection and interpretation of temporal loading metrics and their sensitivity to rainfall representation and aggregation.**

**Question 1.4.** *Methods: L294: "Rainfall temporal loading metrics are implemented in Python based on the definitions provided in the original publications". It strikes me as odd that in a study that focuses on the calculation of metrics, no equations are provided of how metrics are calculated.*

**Response 1.4.**

> We thank the reviewer for this comment and agree that providing explicit equations improves the clarity and transparency of the methodology.
>
> In the original submission, we prioritised reproducible Python implementations because many of the metrics are inherently procedural, relying on ordered steps, thresholds, or windowing operations that are more clearly and precisely expressed in code than in equation form. This approach was intended to avoid ambiguity and support reproducible application by other researchers.
>
> That said, we recognise that including mathematical expressions aids readability and allows readers to more easily compare metrics conceptually. Based on a suggestion from reviewer 3, we have therefore revised the manuscript to include equations for the final recommended metrics (rather than all metrics), alongside references to the corresponding code implementation. This balances analytical clarity with reproducibility, while avoiding duplication for metrics whose definitions are inherently procedural.

**Question 1.5.** *Results and discussion: 4.1 Sensitivity of temporal aggregation: - the findings in this subsection on the effects of temporal aggregation are rather obvious, namely that values of peak intensities are particularly sensitive to temporal aggregation. It doesn't really seem worth analyzing and reporting?*

**Response 1.5.**

> We agree that it is intuitive that peak intensity metrics are sensitive to temporal aggregation. We certainly do not want to claim "surprise" where there isn't any. However, this subsection evaluates the sensitivity of a broad range of temporal loading metrics, many of which are more complex and for which the effects of aggregation are far less predictable. The results presented in Figures 5 and 6 provide quantitative evidence of how different metrics respond to aggregation, revealing substantial variability in sensitivity that cannot be inferred a priori.
>
> Even for metrics where sensitivity might be anticipated, we believe documenting the magnitude and consistency of this effect across events and metrics is important. Such evidence provides a reference against which methodological choices can be evaluated and supports more informed selection of metrics in future studies. In the revised manuscript, we will include a statement that explains that the strong sensitivity of peak intensity-related metrics to temporal aggregation is intuitive/expected. We will also highlight that it is

valuable to identify and contrast the magnitude of sensitivity and direction of change for specific metrics, especially the more complex ones.

**Question 1.6.** *Results and discussion: 4.3 Metric redundancy: the same comment applies here: is it really a new insight that metrics related to time of the peak are related, and similar for mass distribution etc.*

**Response 1.6.**

It is indeed reasonable to expect that metrics which quantify similar aspects of temporal structure, such as the timing of peak intensity, may exhibit some degree of correlation. However, the assertion that redundancy within 'categories' is obvious presupposes the existence of those categories. One of the central contributions of this paper is the development of a coherent categorisation of temporal loading metrics, which enables these relationships, and their limitations, to be clearly identified and interpreted. The updated manuscript will articulate this more clearly.

While some metrics fall into categories quite obviously (e.g., time-to-peak being a peak timing metric), for other metrics this is much less clear (e.g., $skew_p$ being a peak timing metric). Therefore, this manuscript also provides a systematic testing of assumptions about metric behaviour, something which we found to be lacking in the literature. For instance, our empirical analysis allowed us to classify $skew_p$ in the peak timing category, and furthermore provided a clear assessment of the extent to which $skew_p$ is correlated with time-to-peak. This is an invaluable output for anyone wishing to apply either of these metrics, and/or to compare their results to previous studies using either metric.

Additionally, Section 4.3 offers clear evidence that peak timing and mass timing metrics are not equivalent. While this distinction may appear intuitive, we argue that it is frequently obscured in the literature, where broad summary terms such as "front-loaded" are used interchangeably to describe results derived from both peak timing and mass timing metrics. This practice risks treating fundamentally different aspects of temporal structure as equivalent. Explicitly demonstrating and quantifying this distinction is therefore a key motivation for, and outcome of, the systematic metric evaluation presented here. Section 4.3 in the manuscript has been revised to makes this point more clearly.

**Question 1.7.** *5. Recommendations "A central contribution of this paper is the argument that analyses of rainfall temporal loading must begin with a clear definition of the aspects most relevant to the research question or application." This seems like a rather generic and common sense argument. Which comes back to the earlier comment that the new insight provided by this study is not very clear."*

**Response 1.7.**

We agree that, as a general principle, the idea that analyses of rainfall temporal loading should begin with a clear definition of the relevant aspects may appear self-evident. However, the motivation for this paper arises from the observation that, in practice, this step is often not made explicit in the literature. Instead, different metrics are frequently applied, compared, or summarised under common descriptive terms without a clear articulation of the specific aspect of temporal structure they are intended to represent.

The aim of this study is therefore not to argue that such clarity is desirable in principle, but to provide a structured framework through which it can be achieved in practice. By systematically reviewing existing metrics, grouping them according to the aspects of temporal loading they quantify, and empirically evaluating their behaviour, sensitivity, and redundancy, the paper offers a concrete basis for making these definitions explicit and defensible.

In light of this, we agree that the sentence quoted by the reviewer could be misinterpreted as positioning a common-sense principle as a central contribution. We have therefore revised the manuscript to clarify that the paper is *built* on the premise that analyses should begin with a clear definition of the relevant aspects of temporal loading, and that a central contribution of the study is the provision of a framework that enables such definitions to be made more consistently and transparently in future work. The revised text reads as follows:

**"This paper adopts the premise that analyses of rainfall temporal loading should begin with a clear definition of the aspect(s) most relevant to the research question or application. Building on this premise, a central contribution of this work is the development of a conceptual framework that supports explicit, consistent, and reproducible metric selection. Rather than prescribing a single universal metric, we provide a structured approach to choosing metrics that are aligned with the intended interpretation."**

Across the previous three comments (1.5, 1.6, 1.7) it's clear that we have not done enough to motivate the need and importance of establishing a coherent categorisation of metrics that can aid researchers in choosing proper metrics for their analyses. To correct for this problem generally, in the revised manuscript we add some new text on this objective to the end of the Introduction. After establishing the research gap, and the four research questions we add the following text: **"Based on the findings of the empirical analyses related to the four research questions, we will provide a new conceptual categorisation of the various aspects of temporal loading. This aims to provide a new, conceptual framework to assist researchers in interpreting and comparing results from existing research, and to aid the meaningful selection of metrics in future research.**

**Question 1.8.** *6. Summary and conclusions This section should preferably restrict itself to presenting the conclusions. A short summary is already provided in the abstract of the paper.*

**Response 1.8.**

> We agree that the Conclusions section was too wordy and lengthy. This section has now been considerably shortened. This includes movement of some of the sections of text to earlier parts of the manuscript (e.g., as suggested by Reviewer 2, the section on the limitations of sMAPE has been moved to the Methods section). Additionally, in the revised manuscript, unnecessary repetition of earlier details have been omitted.

---

## Author Comment (AC2)

**Reviewer 2 - Anonymous**

The manuscript addresses a common issue when comparing different publications regarding rainfall events and their characteristics, namely that different authors and research groups use a wide range of metrics to characterize rainfall events. Therefore, this analysis, comparing and evaluating a range of metrics used in prior publications is a useful tool for researchers to interpret and compare results from various papers, as well as for authors and researchers to select meaningful metrics for their analysis in future work. As such this represents a valuable contribution to the field of rainfall event analysis.

However, the current manuscript contains certain limitations that require further analysis prior to publication. The decision to use 10 time points when calculating/representing dimensionless mass curves (DMCs) to compare the effect normalisation has on metrics seems flawed and represents a seemingly unnecessary own goal. Using 10 time points aggregates and summarizes the event time series, hence it is unsurprising that DMCs results for metrics that are sensitive to time discretization and aggregation show significant differences compared to the original time series data. Therefore, in its current state, the tests which are meant to examine whether normalization affects metrics seem to be flawed, as the method used both normalizes and aggregates the data. I strongly recommend that the effect of normalization be tested independently of any aggregation, by using the normalized, dimensionless mass curves at their original 5-minute resolution.

**General reply - Reviewer 2**

> Thank you for taking the time to review the manuscript and for your constructive comments. We are pleased that you consider it a useful resource for both evaluating existing research and informing future studies. We have carefully considered your comments regarding the limitations associated with using DMCs interpolated to ten data points. We acknowledge that differences observed when computing metrics on DMCs rather than raw rainfall time series reflect the combined effects of double normalisation and interpolation to ten points. However, we do not view this as a methodological "own goal", but rather as a deliberate methodological choice that reflects the widespread use of DMCs processed in this manner within the literature. DMCs are typically employed to enable comparison or aggregation across events of differing durations, magnitudes, and numbers of observations, for which interpolation to a common number of time steps is a fundamental step. Our literature review suggests that 10 timesteps is a frequently selected choice. We recognise that additional context on the development and typical use of DMCs would help clarify this processing decision, and we have therefore expanded the discussion in Section 2.2.
>
> While we have defended the decision to focus primarily on metric performance calculated on DMCs interpolated to ten data points due to it being common practice, we share the reviewer's curiosity regarding the influence of the normalisation step itself. Accordingly, we have run the same experiments as in the original version of the manuscript but without the aggregation to 10 time steps. These additional analyses are based

on events that have been double normalised but retain their original number of data points, thus isolating the effect of normalisation as requested by the reviewer. The revised version of the manuscript will include an extra appendix presenting additional versions of Figures 7 and 8, accompanied by text describing how these results differ from the original figures.

We also recognise that aspects of our terminology contributed to confusion on this point. In particular, we variously referred to the construction of DMCs as 'normalisation' or 'double normalisation', and to the resulting profiles as 'double normalised' or 'normalised' representations. This was intended as shorthand to avoid repeated use of the DMC acronym, which may be unfamiliar to some readers. However, to improve clarity, we have revised the language throughout the manuscript to more explicitly describe the DMC construction process.

**Question 2.1.** *Line 175: "based on the timing of the peak" intensity? While the context in the text makes it clear that you refer to peak intensity, it is worth being explicit and adopting this phrase here and throughout to avoid confusion with peak volume.*

**Response 2.1.**

Thank you for highlighting this. We agree the statement could be made more clear, and have adjusted the text accordingly.

**Question 2.2.** *Figure 1: The labels in the figure do not match those in the caption (b and c seem to be flipped)*

**Response 2.2.**

Thank you for highlighting this error. The figure has been updated and now shows the correct labels.

**Question 2.3.** *Line 186: I believe there is an opportunity to make this statement more forceful as categorical classifications do mask finer differences between events. From a design rainfall point of view, categorisation may mask other more relevant characteristics such as peak intensities across a range of relevant durations.*

**Response 2.3.**

Thank you for this comment. As suggested we have amended this paragraph to add an extra sentence more strongly highlighting the limitations of classification metrics.

The paragraph now reads: **Classification metrics assign discrete labels to rainfall events. Events are often categorised based on the fraction of the storm containing the greatest rainfall, typically a third**

(..), quarter (..), or fifth (..). While classification metrics offer an intuitive summary of storm structure, their categorical nature masks within-class variability. Events assigned to the same class may differ substantially in peak intensity, duration, or short-timescale concentration, despite sharing a similar overall classification. Furthermore, the classification assigned depends heavily on implementation choices. Firstly, how events are divided into fractions. Secondly, which aspect of rainfall is focused on. The initial third-based classification system was based on the timing of the peak, whereas more contemporary research has based classification on the third containing the highest total rainfall (Liang et al., 2023) or the third containing a summary statistic, e.g. the centre of gravity or $D_{50}$ (see Sect. 2.3.2)

**Question 2.4.** *Line 276: This exclusion is a shame as event definition is probably the single most important factor in determining the ultimate event profile which is being analysed, and while peak intensity is unlikely to be affected by different start/end times, the timing, position, and relative magnitude of the peak intensity is likely to be affected, as will event metrics. Given the importance of this, a lengthier discussion is suggested*

**Response 2.4.**

We agree with the reviewer that event definition is a critically important factor in shaping rainfall event profiles and can substantially influence the timing, position, and relative magnitude of peak intensity, as well as derived temporal loading metrics. We also agree that different choices of event start and end times have the potential to affect many of the metrics analysed in this study.

The decision not to explicitly explore the sensitivity of temporal loading metrics to event definition was therefore not based on a lack of importance, but on considerations of scope. The influence of event boundary definition has been previously examined in the literature, with multiple studies demonstrating its effects on rainfall intensity, duration, depth, and inferred event characteristics. These studies collectively show that event definition alone can materially alter conclusions drawn from rainfall analyses.

In contrast, the focus of this paper is on a set of methodological choices that have received comparatively little systematic attention, namely the definition of the analytical objective (i.e. which aspect of temporal loading is of interest), the selection of metrics used to represent that objective, and the sensitivity of those metrics to rainfall processing choices such as temporal aggregation and normalisation. Expanding the analysis to include multiple alternative event definitions would have substantially increased the dimensionality of the problem and risked obscuring the specific effects this study seeks to isolate.

To address the reviewer's concern, we have expanded the discussion in the manuscript so the paragraph now reads as follows:

> To ensure event independence, we extract events using a minimum inter-event time (MIT) threshold (Restrepo-Posada and Eagleson, 1982; Molina-Sanchis et al., 2016). An 'event' thus constitutes any rainfall separated by at least 11 hours of rain-free conditions, following practice in several Danish hydrological studies (Gregersen et al., 2013; Thomassen et al., 2023). This approach ensures that each event begins and ends with non-zero rainfall. The choice of MIT has been shown to play an important role in determining both the number and properties of rainfall events identified (Dunkerley, 2008). In this study, event definition is treated as a fixed preprocessing choice rather than a variable of investigation, reflecting a deliberate scoping decision. While the delineation of event boundaries can influence the timing and relative prominence of peak intensity, and hence derived temporal loading metrics, its effects have been examined in several previous studies, e.g. Dunkerley (2008, 2010, 2015); Wang et al. (2019); Freitas et al. (2020); Molina-Sanchis et al. (2016); Haile et al. (2011); Medina-Cobo et al. (2016); Meier et al. (2016). In contrast, this study focuses on methodological choices that have received less systematic attention, namely the selection and interpretation of temporal loading metrics and their sensitivity to rainfall representation and aggregation.

**Question 2.5.** *Line 287: Was the impact of using 10 time points examined, say, against using the DMC calculated using the original 5-min resolution? In my experience the number of points used can significantly alter summary metrics if they are calculated using this data rather than on the original measurement intervals*

**Response 2.5.**

> This point relates directly to the broader concern addressed in our response to the general comments regarding the use of DMCs interpolated to ten time points. As discussed there, differences observed when calculating metrics on DMCs reflect the combined effects of double normalisation and interpolation, both of which are intrinsic to how DMCs are commonly constructed and used in the literature.
>
> In the original manuscript, we did not explicitly isolate the effect of interpolation by comparing ten-point DMCs against double-normalised events retained at their native 5-minute resolution. In response to this comment, we have now undertaken this additional analysis and present the results in a new appendix, including supplementary versions of Figures 7 and 8. These allow the specific influence of the interpolation step on metric behaviour to be assessed and are discussed briefly in the accompanying text.

**Question 2.6.** *Line 345: Continuing the point made above, this seems like an issue that could have been solved using a number of time points that is divisible by 4.*

**Response 2.6.**

We thank the reviewer for this suggestion. As discussed in our broader response on the choice to interpolate DMCs to ten points, this decision was made to provide a standardised representation across events of different lengths, magnitudes, and numbers of observations. We specifically chose ten points to follow common implementations of DMCs; in our review of the literature, we have not seen DMCs constructed with only four points. While using a number of points divisible by four would address the calculation of the 'Fraction. in Q1–4' metrics, it would not resolve similar issues for other metrics, such as those based on fifths or other fractional divisions. Creating multiple custom versions of DMCs to suit different metrics would defeat the purpose of having a consistent, generalisable implementation. Hence, interpolation to a shared number of points remains the most consistent and practical approach. This rationale is now clarified in the revised manuscript in Section 2.2.

**Question 2.7.** *Section 4.2: Again, the decision to limit DMCs to 10 points is shown to strongly influence results, I suspect that several of the discrepancies arise more as a result of the use of 10 points than of the normalisation, after all, events can be normalised and presented at their original resolution, which would smooth out the histograms shown in Figure 8.*

**Response 2.7.**

We thank the reviewer for this observation. As noted in our broader response on the use of DMCs interpolated to ten points, it is correct that the differences observed reflect the combined effects of double normalisation and interpolation. To clarify the specific influence of using ten points, we have included additional analyses in a new appendix, where DMCs are double normalised but retain their original 5-minute resolution. These supplementary figures allow the smoothing effect of interpolation on the histograms in Figure 8 to be directly assessed.

**Question 2.8.** *Figure 7: The text in the image is small and difficult to read.*

**Response 2.8.**

Thank you for highlighting this. We have reformatted the figure so that the text is larger and more accessible to readers.

**Question 2.9.** *Line 338: The effects discussed in this and the following paragraph are compelling.*

**Response 2.9.**

> We assume that this comment is intended to refer to L438, rather than L338. If so, we thank the reviewer for the positive comment and are pleased that the discussion of DMC transformations and their implications is found to be compelling.

**Question 2.10.** *Figure 9: As with Figure 7, the text is small and difficult to read.*

> **Response 2.10.**

> Thank you for highlighting this. We have reformatted the figure so that the text is larger and more accessible to readers.

**Question 2.11.** *Line 475: Indeed, it is easy to imagine cases where peak intensities are timed quite differently from the storm's centre of mass.*

> **Response 2.11.**

> Thank you. We agree, but also think that this distinction is often lost in the broader literature, when summary terms such as "front-loaded" are used in place of more specific descriptors. This can lead to misinterpretation, where metrics measuring different aspects are treated as equivalent because they are subsequently described using the same summary term. Highlighting this distinction is therefore one of the key motivations for our systematic metric evaluation.

**Question 2.12.** *Recommendations: The recommended metrics should be highlighted and listed within the text. This is probably the most important contribution of this manuscript, and it should be made more clearly.*

> **Response 2.12.**

> Thank you for this suggestion. We agree and have added the following sentence to the first paragraph of the recommendations: **"The recommended metrics are: the 4th with most rainfall and $D_{50}$ (mass timing); the peak-position-ratio (peak timing); Gini coefficient (magnitude concentration); Temporal standard deviation (temporal concentration); and the event dry ratio (intermittency)."**
>
> In response to reviewer 3's question 3.4, we further add the equations for the recommended metrics in the same section.

**Question 2.13.** *Summary and conclusions: This section is currently a bit long and repetitive, and some aspects would be better discussed elsewhere (for example, the sMAPE limitations could be moved to the Methods section)*

**Response 2.13.**

> The Summary and Conclusions section has been considerably shortened. As helpfully suggested here, the discussion of the sMAPE limitations has been moved to the Methods section. Additionally, unnecessary repetition of earlier details have now been omitted.

**Question 2.14.** *Appendix A: The exclusion of the term "hyetograph" is surprising given that a sizeable number of papers refer to rainfall distributions with this term*

**Response 2.14.**

> We thank the reviewer for this comment. The term 'hyetograph' was not explicitly included in the formal Boolean search because it is primarily used in the literature to describe a rainfall input to hydrological or hydraulic models, rather than as the object of study in investigations of temporal rainfall characteristics. In contrast, terms such as 'rainfall temporal profile', 'temporal loading', and 'intensity profile' are more likely to appear in papers that explicitly focus on the temporal structure of rainfall events, which was the aim of our review. After reading your concern, we tried conducting some ad-hoc searches using "hyetograph" to verify that no major studies were omitted. We were satisfied with the results, and therefore consider that the literature review remains comprehensive and representative of the field.

---

## Author Comment (AC3)

**Reviewer 3 - Anonymous**

The authors review and compare metrics for the "temporal loading" of rainfall events assessment i.e. the temporal distribution of the precipitation intensity within rainfall events. They carry out an extensive literature review and identify 52 metrics which have been applied to describe this temporal loading of events. They categorise these metrics into classification metrics, summary statistics and intermittency metrics. They apply the metrics for a large data set of rainfall events observed in Denmark and compare them regarding sensitivity to temporal resolution and redundancy. Finally, the recommend metrics for the characterisation of special aspects of temporal loading.

The authors have invested much time in analysing the large amount of literature and the many metrics. The paper is not innovative regarding new methods or better description of hydrologic phenomena. However, it is useful for the selection of appropriate metrics for own analyses, so it should be published after a revision.

It is somehow a mixture between a review paper, a research paper and a technical note. For a technical note it is too long for a pure review it contains too much calculations. Maybe it can be shortened to become a technical note. I am not sure if that is feasible but it would allow to concentrate on the most important results and the final recommendations. Also, it would avoid some lengthy descriptions which are partly tiresome to read. A few suggestions for shortening are given below.

**General reply - Reviewer 3**

> Thank you for taking the time to review the manuscript and for your constructive comments. We are pleased that you consider the work useful for researchers seeking to select appropriate metrics.
>
> We acknowledge your point regarding the mixed nature of the manuscript, combining elements of a literature review, research paper and a technical note. However, we firmly believe that this manuscripts strength lies in its structuring around two linked components. Firstly, the literature review establishes how rainfall temporal loading is conceptualised and quantified across different research communities, highlighting substantial variation in terminology and metric usage. And secondly, the large-scale empirical evaluation of 52 metrics across more than 200,000 rainfall events, which quantifies robustness to aggregation and calculation on dimensionless mass curves, and studies the relationships between metrics. We believe that the novel synthesis offered in the literature review is a key scientific contribution of this work, and provides the necessary context for interpretation of the empirical evaluation of the metrics which follows. We believe that converting this analysis to a technical note, and removing the literature review, would render the empirical analysis difficult to interpret and reduce its relevance for cross-study comparison. For this reason, we maintain that the Research Article format remains appropriate, while recognising the need for greater concision and clarity.

To address these concerns, we have substantially shortened the literature review from 7.5 to 5.5 pages (approximately a 30% reduction), with the review methodology, Figure 3, and Table 1 moved to an appendix. The remaining text has been edited to reduce verbosity. We believe these revisions improve the focus and accessibility of the manuscript while retaining the full conceptual and empirical contribution of the work: namely, providing a structured framework for understanding, interpreting, and comparing rainfall temporal loading metrics, and supporting deliberate metric selection in future studies.

**Question 3.1.** *1. Introduction: The paper discusses temporal loadings of fixed storms based on the Eulerian view of the storms. The authors should mention, that both Euler (fixed storm) and Lagrange view (moving storm e.g. described using radar data) would be possible to calculate such metrics. In addition, the metrics analysed purely describe 1D temporal loadings. The natural storms however have space-time dimensions, so 2D temporal loadings may also be possible.*

> **Response 3.1.**

We thank the reviewer for highlighting this perspective. Our study adopts an Eulerian view, analysing rainfall events at fixed locations and considering only 1D temporal loading. We recognise that natural storms have intrinsic space–time structure, and that a Lagrangian approach, in which moving storm systems are tracked, represents an alternative method for characterising rainfall events and their temporal loading. In principle, similar metrics could be calculated to analyse the temporal profile of a moving storm as a whole; however, our literature review did not identify any established metrics that achieve this. Accordingly, we constrained our study to fixed-location rainfall. We have amended the Introduction's opening paragraph to explicitly note this scope:

**In this study, we focus on rainfall measured at fixed locations (Eulerian perspective), which provides a simple, tractable representation of temporal variability. Lagrangian approaches, which track the evolution of moving storms across space, are possible in principle, but methods for quantifying temporal loading in this context are not yet established. We therefore use the term 'event temporal loading' to describe internal variability in intensity over the course of a storm, as considered from a fixed location.**

**Question 3.2.** *2. Literature review: The details how the literature review has been carried out could be omitted if the paper is to be shortened.*

> **Response 3.2.**

> Thank you for this suggestion. We agreed that the methodology of the literature review would be better placed in an Appendix, and have now moved it there.

**Question 3.3.** *3. Statistics regarding the literature review e.g. table 1 or fig. 3 are not really necessary if the paper is to be shortened.*

**Response 3.3.**

> This is a useful suggestion. In response, we have decided that Table 1 and Figure 3 will be relocated to an Appendix in the revised manuscript. The (now shortened) literature review, directs readers to go to the appendix if they are interested in summary overviews of the reviewed literature.

**Question 3.4.** *4. Recommendations: This should be one of the main outcomes. The equations for the recommended metrics should be given.*

**Response 3.4.**

> As suggested, we have amended the Recommendations section to include equations for the recommended metrics.

**Question 3.5.** *5. References: The references should be listed beginning with the last name of the first author.*

**Response 3.5.**

> The reviewer is correct that the reference list was incorrectly formatted. This has now been updated so that the last name of the authors is listed first.